# Mendelian randomization study of self-reported long sleep duration, short sleep duration, and insomnia and cognitive function

**Yunyun Guo**[ID]*

Ageing Epidemiology Research Unit, School of Public Health, Imperial College London, London, United Kingdom

* y.guo22@imperial.ac.uk

## Abstract

### Background and aims

Causal relationship between sleep duration and cognitive function remains unclear. This study used a two-sample Mendelian randomization (MR) analysis to assess the causal relationship between self-reported short sleep duration, insomnia and long sleep duration and cognitive function.

### Methods

A total of 26 single nucleotide polymorphisms (SNPs) associated with short sleep duration, 240 associated with insomnia, and 7 associated with long sleep duration were extracted from a genome-wide association study primarily based on European ancestry, to be used as instrumental variables. Summary-level statistics were obtained from the Dementia genome-wide association studies database. MR estimation was performed using the inverse variance weighted (IVW) method as the primary method, supplemented by MR-Egger regression and weighted median estimator methods. Finally, multiple sensitivity analyses were performed to obtain robust and valid estimates.

### Results

Based on IVW methods, short sleep duration showed a harmful impact on cognitive performance score (beta = −0.15, 95% CI: −0.27 to −0.02, P = 0.02, IVW), fluid intelligence score (beta = −0.38; 95% CI: −0.65 to −0.11; P = 0.006, IVW), memory performance (beta = −0.10, 95% CI: −0.20 to −0.0005; P = 0.04, IVW) and Trail Making (TM) test (TM: interval in trail 2 path, beta = 0.11, 95% CI: 0.01 to 0.21; P = 0.03, IVW; TM: duration to complete trail 2 path, beta = 0.11, 95% CI: 0.002 to 0.22; P = 0.04, IVW). In addition, insomnia was causally associated with Alzheimer's disease (OR = 1.13, 95% CI = 1.02–1.24, p = 0.02, IVW). However, due to the limited number of SNPs

**Data availability statement:** The summary-level data utilized in this investigation were sourced from publicly accessible European pedigree genome-wide association studies (GWAS). Direct links to the specific datasets used are available in S1 Table.

**Funding:** The author(s) received no specific funding for this work.

**Competing interests:** The authors have declared that no competing interests exist.

(n = 7) available as instruments for long sleep duration, there was no strong evidence to support a causal effect of long sleep duration on cognitive outcomes.

## Conclusions

This study suggests self-reported short sleep duration was causally associated with cognitive decline and self-reported insomnia was causally associated with increased risk of Alzheimer's disease in individuals of European ancestry. The evidence of causality between long sleep duration and cognitive function requires further investigation. These results may have implications for public health interventions aimed at reducing the risk of cognitive decline.

---

## 1 Introduction

Appropriate sleep duration has been the focus of public health recommendations [1]. The National Sleep Foundation (NSF) and the American Academy of Sleep Medicine (AASM) recommend that all adults should sleep at least 7 hours per day [2,3]. However, the 2020 U.S. National Survey indicates that more than 25% of adults aged 18 and older sleep less than 7 hours [4]. Despite the adverse health effects of restorative sleep deprivation, sleep problems often go undiagnosed and untreated [5,6]. Although elevated challenges in sleeping may arise due to typical alterations in "sleep architecture," persistent health conditions, or the side effects of medications, older individuals and their healthcare providers should not merely acknowledge sleep issues as an inherent aspect of aging without making efforts to address and alleviate them. A recent study found that adults with sleep problems, particularly insomnia, have significantly higher healthcare utilisation and healthcare spending. The total direct healthcare cost of sleep-related nationally exceeds $90 billion [7]. This may be related to the multiple downstream pathogenic events caused by too much or not enough sleep.

Recently, there has been increasing evidence that long sleep duration, short sleep duration, and insomnia are associated with cognitive function [8,9]. Cognition is a combination of brain processes, including the ability to learn, remember and make judgements. Impaired cognitive function may have a profound effect on an individual's overall health and well-being. This is because cognitive decline ranges from mild cognitive impairment to dementia, which is a decline in ability severe enough to interfere with daily life [10]. Current treatments are not optimal for disorders associated with cognitive decline. Instead, sleep are manageable and preventable condition, so understanding sleep phenotypes has attracted increasing attention as a potential target for preventing cognitive decline.

However, current findings are not entirely consistent regarding the relationship between sleep duration and cognitive impairment. A cross-sectional study that included a community group of more than 2000 individuals aged 40 years or older and assessed cognitive performance using changes in scores on the Brief Mental Status Exercise (MMSE) scale showed that individuals with self-reported sleep duration ≤ 5h, 6h, 8h, and ≥9h had odds ratios (ORs) of 2.14, 1.13, 1.51, and 5.37 for cognitive impairment,

respectively (all P<0.01), which suggests a U-shaped association between sleep duration and cognitive impairment [11]. Keil et al. conducted a cross-sectional study of 826 older adults, which was a retrospective analysis of the Seattle Longitudinal Study [12]. The study used the MMSE and Mattis Dementia Rating Scale (DRS) to measure cognition, thereby exploring the association between self-reported sleep duration and cognitive performance in older adults. After moderating for multiple potential confounders, it was found that while both short sleep duration (<7h/d) and long sleep duration (>7h/d) showed an association with a high risk of cognitive impairment, short sleepers had a significantly increased risk of their cognitive impairment (HR=3.67, P<0.005), whereas the trend of the risk of cognitive impairment was not significant in the long sleepers (HR=1.91, P>0.05) [12]. A prospective longitudinal study included 119 healthy adults aged 55 years or older, subjectively assessed for sleep duration and quality, and underwent magnetic resonance imaging (MRI) testing and neuropsychological assessment every 2 years. After adjusting for multiple confounders, a linear regression relationship was found between short sleep duration and age-related brain atrophy including cerebral grey matter, superior frontal gyrus, and inferior frontal gyrus (all P<0.05), in which each 1h decrease in sleep duration at baseline level (6.7h) was associated with a 0.59% increase in ventricular dilatation (beta=−0.587, P<0.01), which may suggest that short sleepers' overall cognitive levels decreased [13]. Benito-leon et al. included 2751 community-dwelling older adults aged 65 years or older, including 298 short sleepers (<5h/d), 1086 long sleepers (≥9h/d), and 1331 regular sleepers (6–8h/d) to assess cognitive decline using changes in MMSE scores. After 3 years of follow-up, no statistically significant difference in cognitive function was found between short sleepers and regular sleepers (P=0.142), whereas long sleepers showed a decline in cognitive function when compared with regular sleepers (P=0.040), and the difference in cognitive function remained statistically significant after adjusting for relevant confounders (P<0.05) [14].

It is worth noting that apart from the inconsistency of the findings, most of the previous studies were cross-sectional and cohort studies. However, there are some shortcomings in drawing causal inferences with the help of observational studies. First, in assessing the causal relationship between sleep characteristics and cognitive function, different observational studies used different adjustment models and did not fully incorporate possible confounders due to the limitations of the studies themselves. Second, when examining the role of sleep phenotypes on cognitive function, they are susceptible to reverse causality bias. Third, differences in definitions of cognitive function (scores) as well as long sleep duration, short sleep duration and insomnia allow for some heterogeneity between studies. The causal relationship between long sleep duration, short sleep duration, insomnia, and cognitive function is currently unclear. Mendelian randomization (MR) studies may be an effective way to explore this issue.

MR is a novel epidemiological method based on genome-wide association studies (GWAS) that uses genetic variants such as single nucleotide polymorphisms (SNPs) as instrumental variables (IVs) to reveal causal relationships [15]. These genetic variants are equally, randomly and independently distributed when assigned.The greatest advantage of MR is that it avoids the risk of bias associated with observational experiments such as potential confounding, reverse causation, etc., and is to some extent similar to randomised controlled studies [15].

This study investigated the causal relationships between long sleep duration, short sleep duration, insomnia and multiple cognitive outcomes through comprehensive two-sample MR analyses of seven cognitive function tests (including cognitive performance, fluid Intelligence score [FIS], memory performance, Trail Making (TM) test, Symbol Digit Substitution (SDS) test, Pair Matching (PM) test, reaction time (RT)) and four dementia types (including Alzheimer's disease [AD], Lewy body dementia, vascular dementia and frontotemporal dementia) derived from the IEU Open GWAS project. The aim was to genetically elucidate the role of these sleep phenotypes in the development of cognitive decline, thereby contributing to the development of new preventive strategies.

## 2 Methods

### 2.1 Mendelian randomization study design

This study utilized a two-sample MR design employing summary statistics of SNPs as IVs to elucidate causal relationships between self-reported long sleep duration, short sleep duration, insomnia, and cognitive function (see **Fig 1**). The

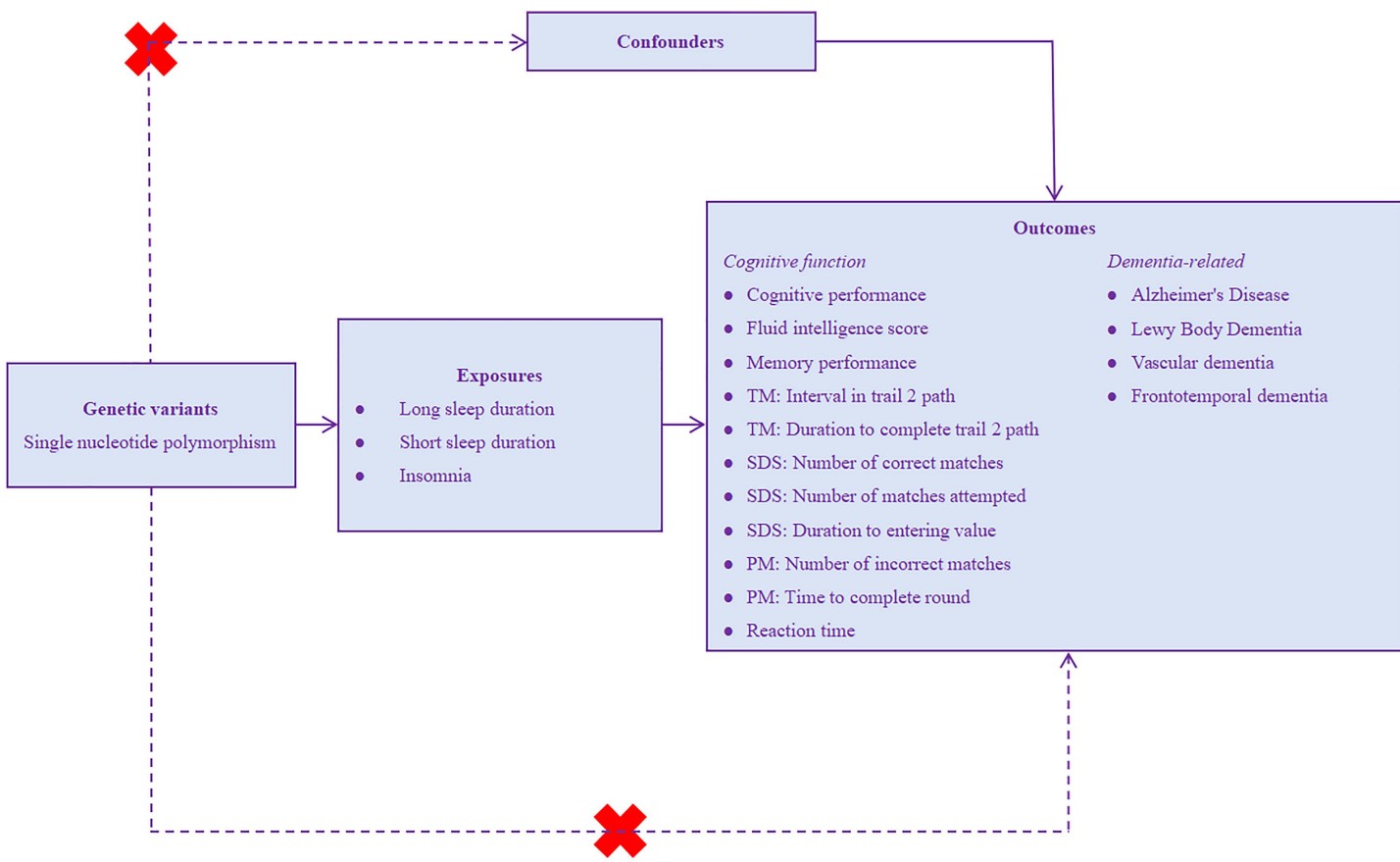

**Fig 1. MR analyses offer a methodological approach for evaluating the causal relationships between exposures, such as self-reported long sleep duration, short sleep duration, and insomnia, and outcomes, encompassing seven cognitive tests and four types of dementia.** The validity of such analyses hinges upon the fulfillment of three pivotal assumptions. Firstly, the genetic variant is strongly associated with self-reported sleep phenotypes. Secondly, it is imperative that the genetic variant remains independent of potential confounding variables. Thirdly, the genetic variant does not have a direct relationship with cognitive function, but affects cognitive function only through pathways that involve these self-reported sleep phenotypes. *Abbreviations*. TM = Trail Making, SDS = Symbol Digit Substitution, PM = Pair Matching.

summary-level data utilized in this investigation were sourced from publicly accessible European pedigree GWAS. The data employed in this MR analysis were drawn from extensive GWAS endeavors, with the original cohorts having received appropriate ethical clearances and obtained informed consent from participants. Consequently, no separate ethical approval was deemed necessary for the present study.

## 2.2 Data sources

**2.2.1 Long sleep duration, short sleep duration, and insomnia.** Through a study analysed genome-wide associations in 446,118 adults of European ancestry, Dashti et al (2019) found 78 loci for sleep duration [16]. Participants in the study were drawn from the UK Biobank study, a prospective cohort study of over 500,000 UK participants. Notably, although the study sleep duration was self-reported, the results were supported by accelerometer-derived estimates. The 78 loci were generally consistent with accelerometer estimates of sleep duration. Participants measured this by wearing a wrist-worn accelerometer for up to 7 days [16]. Relative to a sleep duration of 7–8 h (n = 305,742 controls),

separate GWAS for short sleep (<7 h; n = 106,192 cases) and long sleep (≥9 h; n = 34,184 cases) identified 27 and 8 loci, respectively [16]. Study participants (approximately 500,000) self-reported sleep duration at the baseline assessment. Participants were asked: Approximately how many hours of sleep do you get every 24 hours (please include naps)? Responses were given in hourly increments. Normal sleep duration is 7 or 8 hours. Sleeping 6 hours or less per 24 hours was defined as short sleep duration. Sleeping 9 hours or more per 24 hours was defined as long sleep duration. These thresholds were chosen based on previous epidemiological and clinical studies that have shown that both less than 6 hours and more than 9 hours of sleep are associated with adverse health outcomes, including increased risk of cardiovascular disease, metabolic dysfunction, and cognitive decline [3,16]. Extreme responses of less than 3 hours or more than 18 hours were excluded from the study and responses that were not known or unwilling to answer were set as missing [16].

On the other hand, a GWAS analysis of insomnia by Jansen et al (2019) identified novel risk loci and functional pathways, with original data from UK Biobank and 23andMe, both prospective cohort studies [17]. By combining data collected by UK Biobank version 2 (n = 386,533) and 23andMe (n = 944,477), the study achieved a sizeable sample size of 1,331,010 individuals. The study was meta-analysed for insomnia GWAS in the UK Biobank and 23andMe cohorts [17]. In the UK Biobank cohort, cases of insomnia were identified by self-report of the question: Do you have trouble falling asleep at night or waking up in the middle of the night with responses never/rarely, sometimes, usually, prefer not to answer. Insomnia was defined as the 'usually' response to this question. In the 23andMe cohort, the classification of individuals as insomnia cases or controls was assessed by scoring responses to seven conceptual sleep-related questionnaires [17]. (see **Table 1** for basic information on SNPs)

**2.2.2 Cognitive function and four types of dementia.** The present study presents a comprehensive overview of the cognitive function tests utilized, as outlined in **Table 2**. These assessments encompass a spectrum of cognitive domains, including but not limited to the cognitive performance, FIS, memory performance, TM test, SDS test, PM test, and RT [18–28]. Each test comprises distinct subtests, scoring mechanisms, and participant demographics, all of which are meticulously delineated in **Table 2** for clarity. This study also used four types of dementia from AD Neuroimaging Initiative dataset and Finnish GWAS database: AD (5,918 cases: 212,874 controls); Lewy body dementia (2,591 cases: 4,027 controls); vascular dementia (194 cases: 211,300 controls); and frontotemporal dementia (515 cases: 2,509 controls). For more detailed information refer to S1 Table.

## 2.3 Instrumental variable selection

The flowchart of the study is shown in **Fig 2**. Briefly, sleep phenotype serves as the exposure and cognitive function serves as the outcome.

Initially, SNPs were derived from aggregated GWAS datasets pertaining to sleep phenotypes. SNPs surpassing the genome-wide statistical significance threshold ($p < 5 \times 10^{-8}$) were identified as IVs and subjected to analysis, incorporating measures to mitigate potential biases associated with linkage disequilibrium. Specifically, SNPs were pruned based on a stringent threshold of $r^2 < 0.001$ and a genetic distance of 10,000 kb, ensuring their independence and reliability as IVs. Subsequently, data from both sleep phenotype and cognitive function GWAS databases were retrieved, harmonized, and merged to ensure coherence in the effect alleles pertaining to both phenotypes. Palindromic SNPs, characterized by allele

**Table 1. Description of the genetic instruments associated with self-reported short sleep duration, long sleep duration, and insomnia.**

| Sleep Phenotypes | Max No. of SNPs | Max Sample size | Category | Consortium | Population |
|---|---|---|---|---|---|
| Short sleep duration (<7 vs. 7–8 h/day) | 26 | 411,934 | Binary | UK Biobank or MRC-IEU | European |
| Long sleep duration (≥9 vs. 7–8 h/day) | 7 | 339,926 | Binary | UK Biobank or MRC-IEU | European |
| Insomnia | 240 | 1,331,010 | Binary | UK Biobank and 23andMe | European |

**Table 2. Description of cognitive function tests.**

| Phenotype | Description | Test Details | Scoring | Performance Measure | Data Source | Population Size | Mean±SD |
|---|---|---|---|---|---|---|---|
| Cognitive Performance | Direct observation of activities | 7 subtests | Average task performance; Lower scores indicate greater cognitive impairment; Cronbach's α = 0.71 | Total Score | Social Science Genetic Association Consortium Project | N = 257841 | 0 ± 1 |
| FIS | Touch-screen test with verbal and numerical questions | 13 tests | Number of correct answers in 2 minutes; Cronbach's α = 0.62 | Test Score | Online GWAS summary data from the Integrated Epidemiology Unit of the Medical Research Council, UK | N = 149051 | 6.16 ± 2.15 |
| Memory Performance | Matching task on a touch-screen computer | — | Number of errors on matching tasks; Log-transformed | Errors | MRC Integrative Epidemiology Unit | N = 112065 | 4.06 ± 3.23 |
| TM test | Connect circles in ascending order (Parts A and B) | Part A and Part B | Various metrics including interval in trail 2 path and duration to complete trail 2 path | Part A: Interval (s) Part B: Duration (s) | MRC Integrative Epidemiology Unit | N = 105,451 (interval), N = 99,477 (duration) | Part A: 29 Part B: 75 |
| SDS test | Paper-and-pencil test matching symbols to numbers | — | Number of correct matches in 90–120 seconds; Number of matches attempted; Duration to entering value | Correct Matches/ Attempts/ Duration | MRC Integrative Epidemiology Unit | N = 113,106 (matches), N = 113,410 (duration) | Varies |
| PM test | Memorization and matching of card pairs | — | Number of incorrect matches; Time to complete round | Incorrect Matches/ Time | MRC Integrative Epidemiology Unit | N = 462,302 (matches), N = 454,157 (time) | Varies |
| RT | Timed symbol matching test | — | Average reaction time in milliseconds for matching trials | Avg Reaction Time (ms) | Centre for Cognitive Ageing and Cognitive Epidemiology | N = 36,035 | 555.08 ± 112.19 |

pairs such as A/T or G/C, were systematically excluded from the analysis due to their propensity to introduce inconsistencies in strand orientation and allele coding, thereby potentially confounding the interpretation of effect alleles across datasets. To maintain consistency in strand orientation, only IVs derived from the same DNA strand orientation were retained for further analysis. Moreover, to assess potential associations of SNPs associated with sleep phenotypes with other genomic traits at the genome-wide level, a thorough search was conducted within the PhenoScanner database, with a particular emphasis on excluding potential confounders influencing cognitive outcomes. Following this rigorous selection process, the identified SNPs were utilized as IVs in subsequent two-sample MR analyses. Finally, to evaluate the likelihood of weak instrumental bias, F statistics were computed, with a value exceeding 10 indicative of a reduced probability of weak instrumental bias [29].

## 2.4 Statistical analyses

**2.4.1. Primary analysis.** The primary analyses were conducted to investigate the causal relationships between sleep phenotypes and cognitive function through MR approaches. When dealing with phenotypes characterized by a IV, the Wald ratio test was employed to gauge the correlation between the identified IV and each cognitive function [30]. Conversely, for phenotypes incorporating multiple IVs, the Inverse-Variance Weighted (IVW) test was utilized [31]. The IVW method stands as the predominant approach for computing the weighted mean of all IV effect values [31]. This method operates under the assumption that all genetic variants utilized as IVs are valid and are solely linked to cognitive function through the intermediary of the sleep phenotypes, namely, long sleep duration, short sleep duration, and insomnia. Assuming the absence of horizontal pleiotropy, IVW is presumed to furnish unbiased estimates of causal

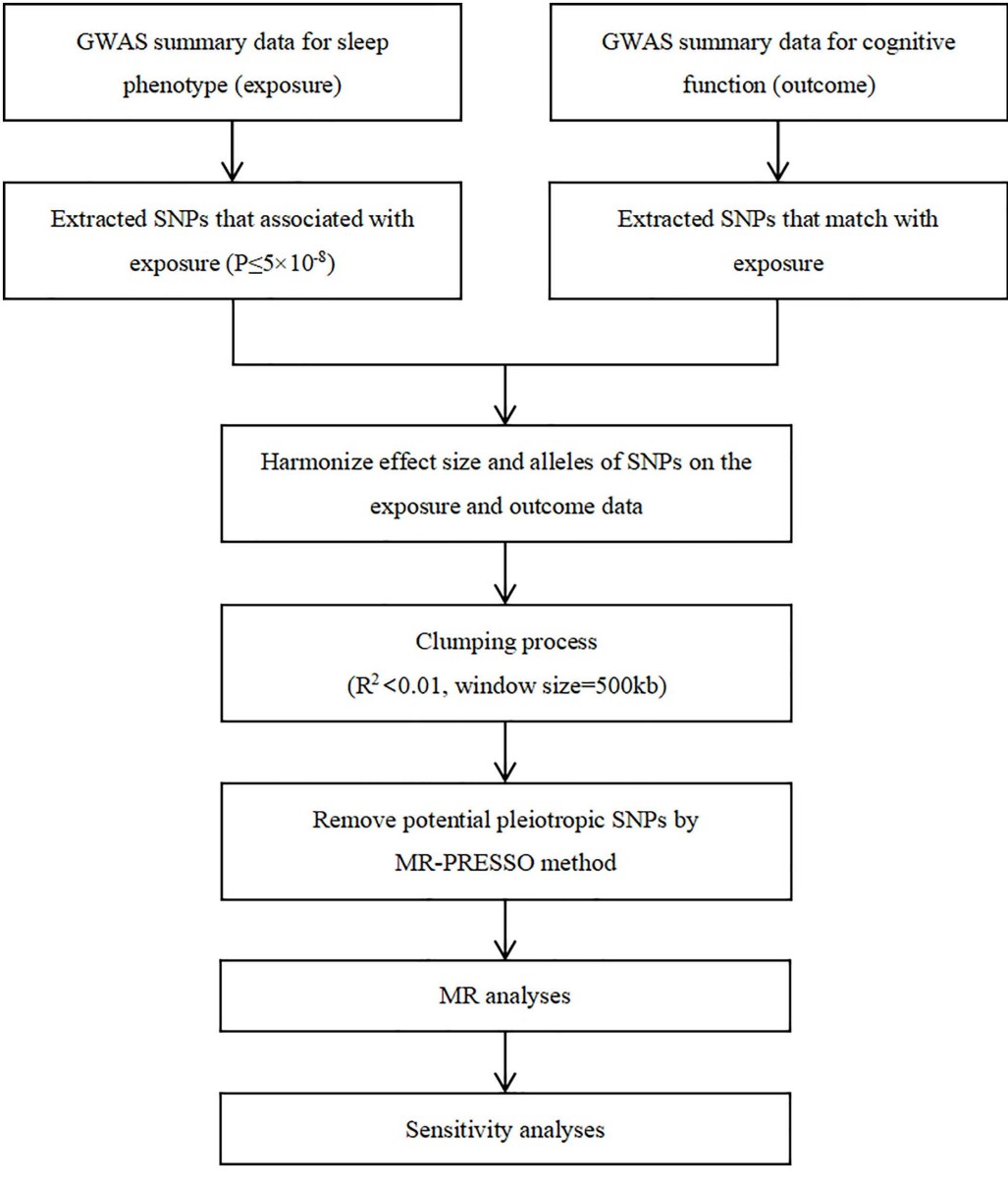

**Fig 2. Study workflow.**

effects. The crux of IVW lies in assigning greater weight to instruments with smaller standard errors, thereby prioritizing more precise estimates.

2.4.2 **Sensitivity analysis.** Numerous sensitivity analyses were conducted to evaluate the robustness of the findings. Initially, two additional methods, namely MR-Egger regression and the weighted median estimator (WME) methods, were utilized in conjunction with the IVW method to refine the estimation outcomes and validate the consistency of MR results. MR-Egger regression operates under the assumption that the strength of the IVs (i.e., the association between genetic variants and the exposure) is independent of their association with the outcome [32]. Conversely, the weighted median estimator assumes that at least 50% of the analytical weight is derived from valid instruments unaffected by

horizontal pleiotropy [33]. Incorporating MR-Egger regression and WME introduces flexibility by relaxing the assumption of absent horizontal pleiotropy in MR. Consequently, these approaches offer enhanced robustness against violations of the assumption that all genetic instruments remain unaffected by pathways other than the sleep phenotype, thereby bolstering confidence in the estimator when consistent results align with the IVW method. Subsequently, a funnel plot was constructed to preliminarily assess heterogeneity; deviation from a funnel-shaped distribution might indicate its presence. Cochran's Q values were then computed to quantitatively evaluate SNP heterogeneity, with heterogeneity deemed present if the associated P-value was less than 0.05 [34,35]. The presence of heterogeneity does not necessarily contravene the fundamental assumptions of MR. In instances where heterogeneity was evident, the random effects IVW method was employed for the MR analysis. Furthermore, a horizontal pleiotropy test was conducted to validate the reliability of the MR analysis outcomes [36], often gauged through the intercept term of the MR-Egger test. An intercept term exceeding P > 0.05 suggests an absence of horizontal pleiotropy; conversely, the presence of horizontal pleiotropy undermines the reliability of the MR analysis results. Additionally, the MR Pleiotropy RESidual Sum and Outlier (MR-PRESSO) test was utilized to identify and adjust for outliers attributable to horizontal pleiotropy using the Outlier-corrected method within the model. Subsequent MR analyses were reiterated after excluding known pleiotropic SNPs. Lastly, leave-one-out analyses were conducted to ascertain whether a single SNP drove the causal effect, thereby further fortifying the robustness of the findings [37].

Statistical analyses were conducted using the TwoSampleMR package (version 0.5.7) within the R environment (version 4.2.2) [38,39]. In order to enhance the robustness of the findings, a Bonferroni correction was implemented to establish the threshold of significance (0.05/45 MR estimates = 0.001). Consequently, outcomes with a p-value < 0.001 were deemed significant, while those falling within the range of 0.001 < p < 0.05 were considered suggestive; the latter category denotes results that did not attain the Bonferroni-corrected significance threshold but did achieve conventional significance at the 0.05 level.

## 3 Results

### 3.1 Instrumental variable selection

The F-statistics for the association of genetic instrumentation with self-reported short sleep duration, insomnia, and long sleep duration ranged from 26 to 67, suggesting that there was little evidence to suggest weak instrument bias in this study. Further screening was carried out according to the IV screening conditions and the final results are presented in S1 Table. According to the ME-Egger intercept test it was stated that the data were not horizontal pleiotropy (p > 0.05) and could be analysed for MR (see S1 Table).

### 3.2 Primary analysis

The findings suggest that short sleep duration may adversely affect multiple domains of cognitive function. MR analysis using an IVW method showed that short sleep duration showed a harmful impact on cognitive performance (beta = −0.15, 95% CI: −0.27 to −0.02, P = 0.02), which serve as a potential test for identifying people in the community with undiagnosed dementia, and their decline reflects a significant reduction in overall cognitive function. Similarly, there was a strong negative correlation between short sleep duration and FIS (beta = −0.38; 95% CI: −0.65 to −0.11; P = 0.006, IVW). The FIS is a physiologically based measure of cognitive ability related to arithmetic speed and reasoning. In terms of memory function, short sleep duration was linked to poorer memory performance (beta = −0.10, 95% CI: −0.20 to −0.0005; P = 0.04, IVW), further underscoring the broad cognitive impact of inadequate sleep (Fig 3).

As shown in Fig 3, performance on the TM test, which assesses executive function and the ability to switch attention, was also negatively affected. Specifically, short sleep duration was associated with a longer interval between steps in the Trail 2 path (beta = 0.11, 95% CI: 0.01 to 0.21, P = 0.03, IVW) and a longer total duration to complete the Trail 2 path (beta = 0.11, 95% CI: 0.002 to 0.22, P = 0.04, IVW), suggesting impairments in attention and task-switching capabilities.

There was insufficient evidence to support a causal relationship between insomnia and performance on cognitive tests, including FIS (Fig 4). In addition, insomnia was causally associated with AD, with a significant odds ratio of 1.13 (95% CI: 1.02 to 1.24, P = 0.02, IVW) (*Fig 6*). However, there was insufficient evidence to support a causal relationship between long sleep duration and every cognitive-related outcome (P > 0.05 for all) (Fig 5 and Fig 6). Considering the limited number of SNPs, the causal relationship between long sleep duration and cognitive function still needs further investigation.

### 3.3 Sensitivity analyses

The MR-Egger and WME approaches complement each other, aligning with the primary IVW methods and thereby enhancing confidence in the findings (refer to S1 Fig). Indications of heterogeneity in cognitive function analyses emerge

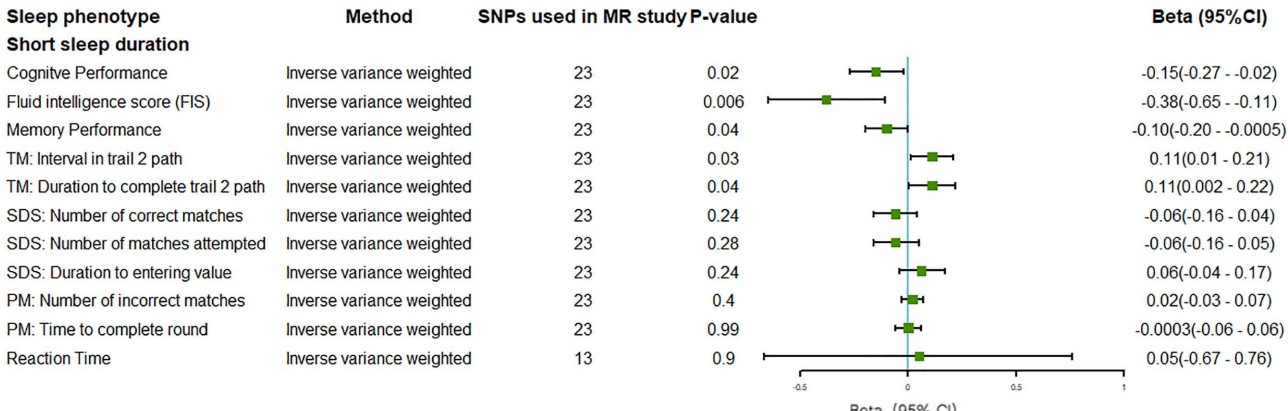

**Fig 3. Forest plot on the associations of self-reported short sleep duration with outcomes related to tests of cognitive function in the main inverse-variance weighted analyses in this MR study.** Beta with the 95% confidence intervals reveal the association estimates with the risk of cognitive decline: self-reported short sleep duration (<7 vs. 7–8 h/day). *TM:Trail Making; SDS:Symbol Digit Substitution; PM:Pair Matching; 95%CI:95% Confidence Interval.*

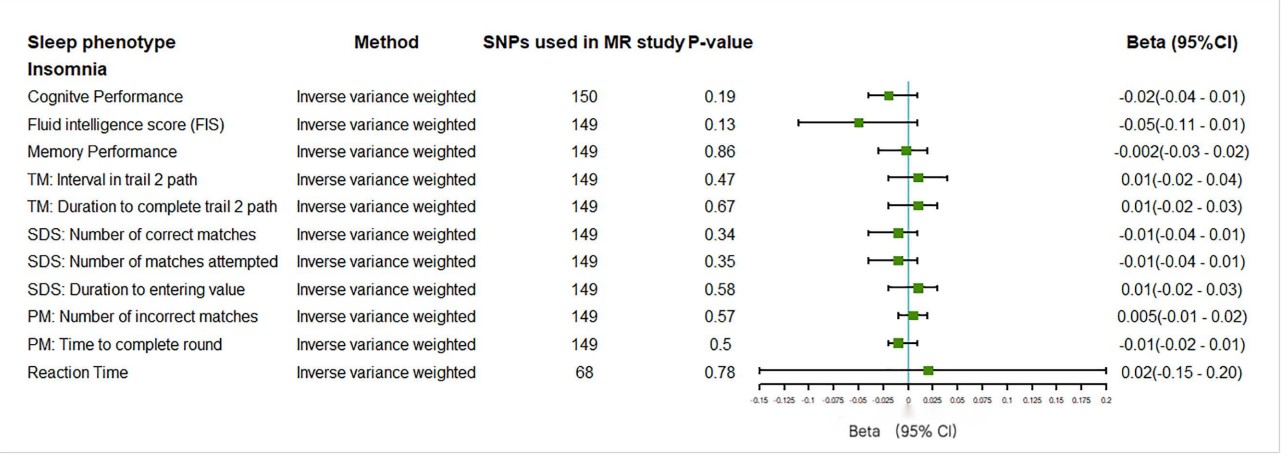

**Fig 4. Forest plot on the associations of self-reported insomnia with outcomes related to tests of cognitive function in the main inverse-variance weighted analyses in this MR study.** Beta with the 95% confidence intervals reveal the association estimates with the risk of cognitive decline: self-reported insomnia. *TM:Trail Making; SDS:Symbol Digit Substitution; PM:Pair Matching; 95%CI:95% Confidence Interval.*

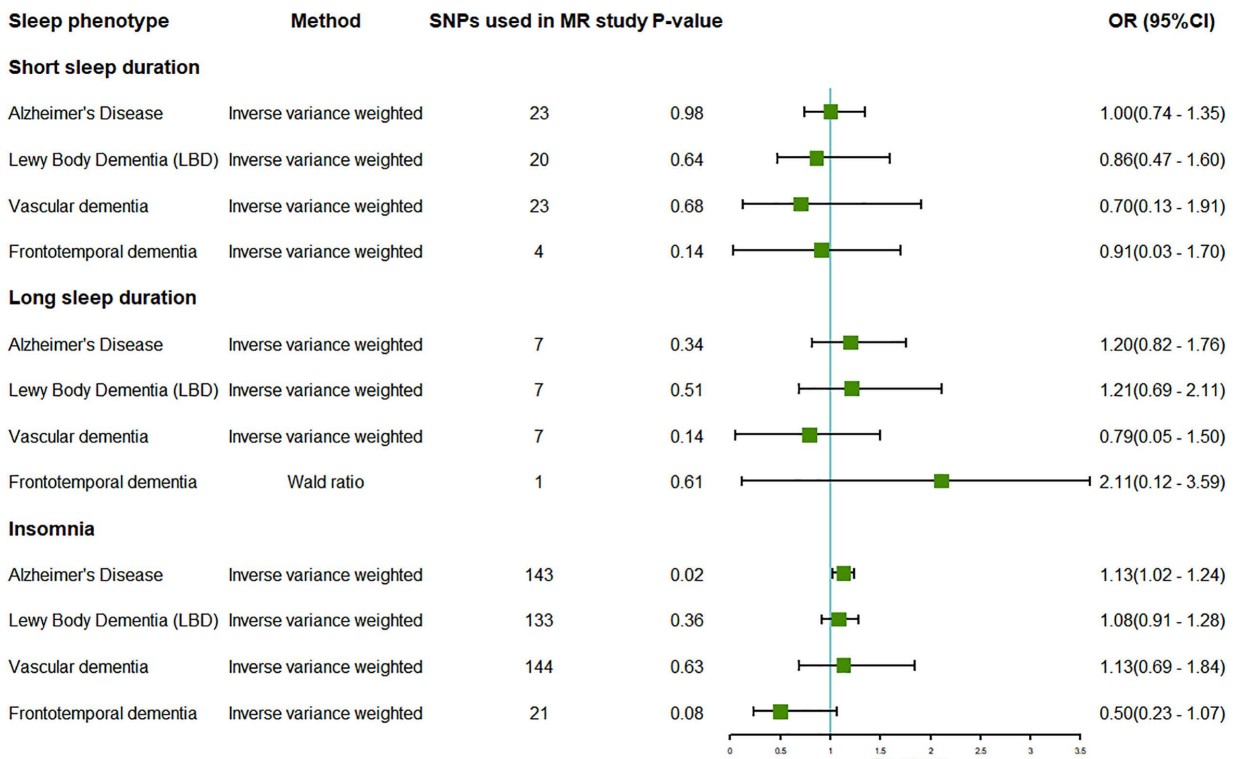

| Sleep phenotype | Method | SNPs used in MR study | P-value | Beta (95%CI) |
|---|---|---|---|---|
| **Long sleep duration** | | | | |
| Cognitve Performance | Inverse variance weighted | 6 | 0.9 | -0.004(-0.08 - 0.07) |
| Fluid intelligence score (FIS) | Inverse variance weighted | 7 | 0.82 | 0.03(-0.19 - 0.24) |
| Memory Performance | Inverse variance weighted | 7 | 0.82 | -0.01(-0.12 - 0.09) |
| TM: Interval in trail 2 path | Inverse variance weighted | 7 | 0.17 | -0.04(-0.11 - 0.02) |
| TM: Duration to complete trail 2 path | Inverse variance weighted | 7 | 0.19 | -0.04(-0.11 - 0.02) |
| SDS: Number of correct matches | Inverse variance weighted | 7 | 0.41 | 0.03(-0.04 - 0.09) |
| SDS: Number of matches attempted | Inverse variance weighted | 7 | 0.27 | 0.03(-0.03 - 0.10) |
| SDS: Duration to entering value | Inverse variance weighted | 7 | 0.24 | -0.04(-0.10 - 0.02) |
| PM: Number of incorrect matches | Inverse variance weighted | 7 | 0.3 | -0.02(-0.05 - 0.01) |
| PM: Time to complete round | Inverse variance weighted | 7 | 0.75 | -0.01(-0.04 - 0.03) |
| Reaction Time | Inverse variance weighted | 3 | 0.71 | -0.23(-1.40 - 0.95) |

**Fig 5. Forest plot on the associations of self-reported long sleep duration with outcomes related to tests of cognitive function in the main inverse-variance weighted analyses in this MR study.** Beta with the 95% confidence intervals reveal the association estimates with the risk of cognitive decline: self-reported long sleep duration (≥9 vs. 7–8 h/day). *TM:Trail Making; SDS:Symbol Digit Substitution; PM:Pair Matching; 95%CI:95% Confidence Interval.*

| Sleep phenotype | Method | SNPs used in MR study | P-value | OR (95%CI) |
|---|---|---|---|---|
| **Short sleep duration** | | | | |
| Alzheimer's Disease | Inverse variance weighted | 23 | 0.98 | 1.00(0.74 - 1.35) |
| Lewy Body Dementia (LBD) | Inverse variance weighted | 20 | 0.64 | 0.86(0.47 - 1.60) |
| Vascular dementia | Inverse variance weighted | 23 | 0.68 | 0.70(0.13 - 1.91) |
| Frontotemporal dementia | Inverse variance weighted | 4 | 0.14 | 0.91(0.03 - 1.70) |
| **Long sleep duration** | | | | |
| Alzheimer's Disease | Inverse variance weighted | 7 | 0.34 | 1.20(0.82 - 1.76) |
| Lewy Body Dementia (LBD) | Inverse variance weighted | 7 | 0.51 | 1.21(0.69 - 2.11) |
| Vascular dementia | Inverse variance weighted | 7 | 0.14 | 0.79(0.05 - 1.50) |
| Frontotemporal dementia | Wald ratio | 1 | 0.61 | 2.11(0.12 - 3.59) |
| **Insomnia** | | | | |
| Alzheimer's Disease | Inverse variance weighted | 143 | 0.02 | 1.13(1.02 - 1.24) |
| Lewy Body Dementia (LBD) | Inverse variance weighted | 133 | 0.36 | 1.08(0.91 - 1.28) |
| Vascular dementia | Inverse variance weighted | 144 | 0.63 | 1.13(0.69 - 1.84) |
| Frontotemporal dementia | Inverse variance weighted | 21 | 0.08 | 0.50(0.23 - 1.07) |

**Fig 6. Forest plot for the associations between self-reported short sleep duration, long sleep duration, and insomnia and outcomes related to tests of cognitive function in the main inverse-variance weighted analyses in this MR study.** OR with the 95% confidence intervals reveal the association estimates with the risk of cognitive decline, respectively: self-reported short sleep duration (<7 vs. 7–8 h/day); self-reported long sleep duration (≥9 vs. 7–8 h/day); self-reported insomnia. *TM:Trail Making; SDS:Symbol Digit Substitution; PM:Pair Matching; OR:Odds ratios; 95%CI:95% Confidence Interval.*

from funnel plots and Cochran's Q tests results, as detailed in S1 Fig and S1 Table. According to the MR-Egger intercept test, there was no evidence of horizontal pleiotropy (all P > 0.05), indicating that the data were suitable for MR analysis (refer to S1 Table for details). As shown in S1 Table, the results of the MR analyses using the outlier-corrected method in MR-PRESSO were compared with those of the original MR analyses, revealing a slightly strengthened causal relationship between short sleep duration and cognitive function, as measured by cognitive performance and FIS. The leave-one-out method did not identify SNPs with a high impact on causal associations (refer to S1 Fig).

## 4 Discussion

Although there have been previous MR studies on sleep duration and cognitive function, the cognitive outcomes assessed in this study were much more comprehensive, encompassing not only four different types of dementia but also seven cognitive tests. This two-sample MR analysis provides causal evidence indicating that genetically determined self-reported short sleep duration is positively associated with an increased risk of cognitive decline. Furthermore, the findings support an association between genetically determined self-reported insomnia and a higher risk of AD. The evidence of causality between long sleep duration and cognitive function remains unclear and requires further investigation.

The results of this study are consistent with previous observational studies of healthy older adults. These studies have found an association between short sleep duration and insomnia and an increased risk of cognitive impairment. Using the English Longitudinal Study of Ageing (ELSA) cohort and the China Health and Retirement Longitudinal Study (CHARLS), Ma Y et al. found that, after adjusting for multiple covariates, overall cognitive scores declined more rapidly among elderly individuals who slept ≤4 hours per night compared with those who reported sleeping 7 hours per night, over 100,000 person-years of follow-up [40]. A meta-analysis that included nine cohorts with a total of 22,187 participants showed that short and long sleep durations increased the risk of cognitive impairment by 34% and 21%, respectively, and that the association between short sleep duration and cognitive impairment was stronger [41]. Yaffe K et al. reported in an 8-year follow-up study of 179,738 male veterans (aged 55 years and older) that individuals with insomnia had a 26% increased risk of developing AD compared with those without insomnia (RR = 1. 26), but did not have a significantly increased risk of vascular dementia or Lewy body dementia [42]. Similarly, Shi L et al. found that insomnia increased the risk of AD but not vascular dementia or all-cause dementia [43].

Reduced sleep duration and insomnia can affect the onset and progression of cognitive impairment and dementia through amyloid β-protein (Aβ) dynamics. Deposition of Aβ is considered a hallmark pathological feature of AD [44]. On the one hand, neuronal firing promotes the rate of Aβ production, which is reduced during slow-wave sleep, resulting in slower or suppressed Aβ production. Reduced sleep following waking leads to increased neuronal activity and frequent discharges that promote Aβ production and accumulation [44,45]. On the other hand, the brain removes metabolic wastes like Aβ more efficiently during slow-wave sleep. Chronic sleep restriction or deprivation can lead to reduced clearance of such harmful substances from the cerebrospinal fluid, thereby increasing Aβ deposition and the formation of amyloid plaques [45,46]. Once amyloid plaques form in the brain they can disrupt sleep-wake function and circadian rhythms. This disruption affects the positive feedback loop that results in poor sleep, which in turn accelerates amyloid deposition and affects sleep-promoting brain regions, creating a vicious cycle of sleep disruption. Animal studies suggest that the rate of Aβ clearance during sleep is twice as high as during wakefulness and that Aβ is significantly deposited during periods of sleep deprivation [47]. Beyond sleep duration, growing evidence suggests that sleep quality plays a critical role in brain health through its effects on neuroinflammation and glymphatic clearance. Disrupted or fragmented sleep has been shown to increase systemic and central nervous system levels of pro-inflammatory cytokines, such as interleukin-6 (IL-6) and tumor necrosis factor-alpha (TNF-α), which can exacerbate neuroinflammatory processes and contribute to neurodegeneration [48]. Inadequate or poor-quality sleep may also impair the function of the glymphatic system—a waste clearance pathway active during sleep—thereby reducing the removal efficiency of amyloid-β and other neurotoxic proteins from the brain [47,49]. These biological mechanisms highlight the importance

of considering both the duration and quality of sleep when evaluating its causal role in cognitive aging and neurodegenerative disease.

Considering the limited number of SNPs (n = 7) available as instruments for long sleep duration, there was no strong evidence to support a causal effect of long sleep duration on cognitive outcomes. Long sleep duration is considered a marker of poor health in older adults. A cross-sectional study that included 4417 cognitively normal older adults aged 65–85 years found no significant difference in Aβ loads between long and normal sleep duration [50]. Longer sleep duration was associated with many other phenotypes associated with aging (e.g., poorer executive functioning, poorer subjective cognition, higher GDS scores, higher BMI), highlighting that the detrimental effects of long sleep duration are generally linked to the aging process itself, rather than to early Aβ pathology [8]. However, some observational studies have found different results. Some studies have suggested a U-shaped relationship between self-reported sleep duration and cognition [51–53]. The results of Li et al.'s study showed that both insufficient and excessive sleep duration were positively associated with poorer performance on subsequent cognitive tasks in the middle-aged and older population [54]. There are two main differences between the Li et al. study and the present analysis. First, Li et al.'s research was based on a prospective cohort design and may be subject to residual confounding. Second, Li et al.'s study differed in its definition of long and short sleep duration. Li et al. dichotomised sleep duration into two categories (≤7 h/24 h and >7 h/24 h). In contrast, the present study defined short sleep duration as 6 hours or less per 24 hours and long sleep duration as 9 hours or more, using 7–8 hours as the reference [54]. A recent longitudinal cohort study enrolling 5,115 participants aged 18–30 years, with sleep duration measured subjectively and objectively using actigraphy, aimed to examine the relationship between sleep duration and quality in early midlife and performance in several cognitive domains in late midlife [55]. This study suggests that the association between sleep quality and cognition may become salient as early as midlife and suggests that sleep quality, rather than quantity, is particularly important for midlife cognitive health [55]. The difference between actigraphy-based and self-reported sleep measures was found to be approximately 0.8 hours [56]. Such methodological differences in sleep assessment may partially account for the inconsistencies across studies.

There have been some previous MR studies on sleep duration and cognitive function. Henry et al. examined the potential effects of sleep duration on cognitive function using pooled statistics from the UK Biobank cohort and the International Alzheimer's Disease Genomics Project [57]. Results from linear MR suggest that a linear increase in sleep duration has a slight negative effect on reaction time and visual memory, but the true association may be nonlinear [57]. The MR study by Wang et al. explored the potential causal relationship between sleep characteristics including sleep duration, insomnia and cognitive impairment [58]. This study reported a negative correlation between self-reported sleep duration and reaction time, suggesting an adverse effect of short sleep duration on cognitive performance. This study did not find sufficient evidence to support a causal relationship between insomnia and cognitive function [58]. Qiu et al. combined data from the National Health and Nutrition Examination Survey (NHANES) and MR analyses to report an inverted U-shaped relationship between sleep duration and cognitive function in older adults in a cross-sectional study, but the MR analyses did not find a causal effect of sleep duration and insomnia on cognitive function [59]. There have also been some MR analyses of the relationship between sleep duration, insomnia, and dementia. MR analyses in the study by Yuan et al. did not find a statistically significant causal relationship between sleep duration and AD, although prospective observational studies in that study suggested that excessive sleep duration may increase the risk of AD [60]. Similarly, Huang et al. did not find any evidence to support a causal relationship between insomnia and AD risk [61].

Five previous MR studies have brought enlightening evidence [57–61]. Based on previous MR analyses, the present study further explored the causal relationship between sleep duration and cognitive functioning by further categorising sleep duration: long sleep duration and short sleep duration. In addition, the present study used insomnia GWAS data based on a larger sample size for exploration. A review of previous MR analyses on insomnia and cognitive functioning found that the study by Wang et al. used insomnia GWAS data from 386,533 individuals of European ancestry recruited from 22 assessment centres in the UK [58]; the study by Huang et al. included insomnia GWAS data from

237,627 individuals of European ancestry in the UK [61]. The present study used combined data collected by UK Biobank (n = 386,533 individuals) and 23andMe (n = 944,477 individuals) for a final sample size of up to 1,331,010 individuals of European ancestry. Insomnia studies with different sample sizes and study sites may yield different conclusions. On the other hand, Qiu et al.'s MR analyses considered only one test of cognitive function [59]. Henry et al. judged cognitive function only by visual memory tests, reaction time and dementia or not [57]. However, the manifestations of cognitive decline are complex. It is difficult to fully reflect changes in cognitive functioning and the presence of dementia using only one or two tests. Compared with the study of Henry et al, Wang et al included more cognitive tests such as cognitive performance, fluid intelligence [58]. Based on previous studies, the present study added more cognitive function tests (seven cognitive tests including memory test, reaction time, cognitive performance and fluid intelligence). At the same time, previous MR studies did not provide more evidence about sleep characteristics and dementia subtypes. Qiu et al.'s MR analysis did not consider any dementia and its subtypes [59]; Henry et al., Yuan et al. and Huang et al. all focused only on AD, the most common form of dementia [57,60,61]; and Wang et al.'s focus fell on AD-related progression scores and changes in brain structure [58].

In the current study, the genetic association analyses yielded some interesting results. Short sleep duration seems to be more strongly associated with certain cognitive indicators, such as fluid intelligence, than insomnia. This may be due to differences in the definitions of short sleep duration and insomnia. In this study, short sleep duration was defined as 6 hours or less of sleep per 24 hours. Shorter sleep duration implies a reduction in slow wave sleep (SWS) and rapid eye movement (REM) sleep. These two sleep stages play different roles in cognitive functioning. SWS, characterised by high-amplitude, low-frequency brain waves, is essential for the hippocampal-neocortical dialogue during memory consolidation [62]. Diminished SWS impairs the reactivation of newly acquired information from the hippocampus and its transfer to long-term storage in the cortex [62]. Thus, SWS is critically associated with declarative memory consolidation and fact-based situational memory [62]. It has been shown that SWS disruption is associated with impaired situational memory and slower information processing, especially after accounting for demographic and lifestyle factors [62]. In contrast, REM sleep is known to promote synaptic plasticity and activate neural circuits involving the prefrontal cortex and striatum to support emotion regulation, attention, and executive function. REM sleep disruption has been shown to impair sustained attention, working memory, and cognitive flexibility, functions that are critically dependent on the integrity of frontal-striatal circuits [63]. The UK Biobank assessment reflects fluid intelligence – which encompasses the ability to process and apply new information, reason abstractly, and perform goal-directed behaviours effectively. Therefore, the observed association between short sleep duration and reduced fluid intelligence scores may reflect cumulative deficits caused by reduced SWS, which affects memory encoding and consolidation, and reduced REM sleep, which affects attention and executive functioning. Despite the small amount of effect observed, MR analyses support that short sleep duration performs equally poorly on other cognitive tests. Specifically, genetically predicted short sleep duration was causally associated with reduced memory performance (beta = –0.10, 95% CI: –0.20 to –0.0005, P = 0.04). This result further highlights the detrimental effects of reduced sleep duration on memory encoding and recall. In addition, impaired executive functioning due to short sleep duration was also reflected in poorer performance on the TM test. Individuals with genetically predicted short sleep duration exhibited longer step intervals (beta = 0.11, 95% CI: 0.01 to 0.21, P = 0.03) and longer total task duration (beta = 0.11, 95% CI: 0.002 to 0.22, P = 0.04), indicating impaired attentional switching and cognitive flexibility. Considering accumulation over time, even mild cognitive decline (e.g., 0.2–0.3 standard deviations) may predict increased risk of functional decline and dementia in later life [52,54,63].

Insomnia is defined as difficulty falling asleep at night or waking up in the middle of the night. Thus, insomniacs may still obtain a normal total sleep duration, albeit with poorer sleep quality. Alternatively, insomniacs may be more conscious of this compensation through planned daytime naps, but chronically sleep-deprived individuals tend to underestimate their cognitive deficits. During planned daytime naps, individuals may not only compensate for REM sleep, but also compensate for reduced SWS by prolonging REM sleep, thus demonstrating resilience to SWS deficits. From this perspective,

some of the functions of SWS sleep may be redistributed during REM sleep stages or compensated to some extent by REM sleep-related processes [63]. Second, sleep duration in this study was supported by accelerometer estimates. Insomnia relies on subjective self-reports, and subjective symptoms are influenced by mood, anxiety, and personal distress, which complicates its connection to objective cognition. In contrast to disease diagnosis, fluid intelligence emphasises speed of information processing, ability to reason abstractly and executive function. Thus, fluid intelligence scores better capture the changes in cognitive functioning that occur as a result of reduced SWS and REM due to short sleep duration.

The present study suggests a close causal link between insomnia and AD, rather than cognitive test scores. There are several plausible explanations for this dissociation. Firstly, cognitive assessment methods provided by the UK Biobank may not readily detect cognitive decline in the preclinical stages of AD. These assessment methods, although suitable for large-scale population studies, may lack the sensitivity to capture subtle or domain-specific cognitive deficits, such as language deficits, disorientation disorders, which are often among the areas susceptible to AD [24,40]. Thus, individuals with early neurodegenerative conditions may still perform normally on these relatively crude tests. Secondly, insomnia may increase long-term risk for AD through pathways that do not lead to immediately measurable cognitive deficits. For example, insomnia is associated with impaired clearance of amyloid β-protein and tau protein by the lymphatic system, systemic inflammation, and neuroendocrine dysregulation, all of which are implicated in the pathogenesis of AD but may precede observable changes in cognitive performance by years or even decades [44,46,49]. Thirdly, individuals with chronic insomnia may also develop compensatory mechanisms or cognitive reserves that temporarily buffer measurable performance deficits, thereby delaying the emergence of impairments on cognitive tests [64]. Finally, in the GWAS included in this study, the diagnosis of AD captures a clinical endpoint, whereas the cognitive performance indicators in the UK Biobank reflect subclinical characteristics. The lack of association between insomnia and cognitive test scores does not necessarily imply a lack of cognitive impact, but should reflect the complexity and time lag between sleep disorders and cognitive decline.

In light of these considerations, it is also important to examine whether structural brain changes—such as brain atrophy—may mediate the relationship between sleep disturbances and cognitive decline. Higher rates of brain atrophy are considered a marker of declining brain health [65–68]. Brain atrophy rates increase with normal aging [65], cognitive decline [66], AD [67], and cardiovascular risk factors [68]. Fjell et al. analysed several large-scale longitudinal MRI datasets and found no evidence of an association between sleep duration and brain atrophy [64]. This result suggests that cognitive decline may not be mediated by sleep duration through increased brain atrophy. However, Fjell et al. noted that the analysis was based on cross-sectional data of brain morphometry and that many covariates potentially influencing the sleep–brain relationship—such as cardiovascular risk factors—were not controlled for [64]. In addition, the study assessed only brain morphometry. Other imaging measures may better capture the relationship between sleep duration and cognitive function, such as hippocampal volume [69] or Aβ accumulation [70]. Another cross-sectional study by Fjell et al. analysing data from 21,000 participants in the UK Biobank, found that sleep durations between 5 and 9 hours per 24-hour period were not associated with smaller hippocampal volumes, whereas shorter and longer sleep durations were [71]. More MR studies on sleep and brain structure, especially the rate of brain atrophy, may compensate for the shortcomings of cross-sectional analyses, thus digging deeper into the intrinsic mechanisms of the relationship between sleep and cognitive function.

This study has several strengths. First, by leveraging the random allocation of alleles at conception, MR analyses reduce bias from residual confounding and reverse causation. SNPs were used as IVs to estimate causal relationships between self-reported sleep phenotypes and cognitive outcomes, under the standard MR assumptions of relevance, independence, and exclusion restriction. Second, most previous studies focused solely on AD, this study included a broader range of dementia types and cognitive function outcomes, including CPT, FI, memory performance, TM test, SDS test, PM test, and RT. In addition, all participants in all GWAS datasets were of European origin, helping to reduce bias due to population stratification. Finally, the three MR methods help to ensure consistency and precision of results. At the same time, several limitations should be acknowledged. First, self-reported sleep duration or insomnia symptoms may

introduce a potential misclassification bias due to memory and reporting errors, which can lead to biased MR analysis estimates towards zero values and underestimation of true causal effects. This bias mainly occurs in cases where there is no difference in measurement error. In this case, the noise introduced by self-reporting weakens the observed association between the genetic IV and the exposure, thereby reducing the strength of the IV. This dilution effect reduces the proportion of exposure variance explained by the IVs, leading to regression dilution bias. As a result, the estimated causal effect of sleep on cognitive or neurological outcomes is attenuated, making it appear weaker or even zero, despite the fact that there may be a genuine underlying association. This issue is particularly important for subjective sleep phenotypes such as insomnia, as individual perceptions can vary widely depending on factors such as mood, stress, and attentional biases. In addition, an increase in exposure misclassification reduces statistical efficacy and widens standard errors, thereby increasing the likelihood of Type II errors. The absence of a flawless method for measuring sleep duration without disrupting daily routines underscores the challenge in this domain [56]. Despite the often robust correlation between actigraphy and polysomnography findings [72], actigraphy tends to overestimate sleep duration, particularly beyond the confines of sleep laboratory environments [72–76]. Notably, genetic correlations with sleep duration appear to be consistent across self-reported and accelerometer-based measures [16], which provides some robustness to the use of self-reported data. In addition, international sleep guidelines continue to rely on self-reported sleep duration to formulate public health recommendations [2,77], which supports their relevance in both clinical and epidemiological contexts. Nonetheless, future research should prioritise the integration of objective sleep measures (e.g., somatokinetic recorders and polysomnography) into large-scale cohort studies to improve measurement accuracy and reduce potential bias in causal inference. Second, the limited number of SNPs used as IVs for long sleep duration likely constrained the statistical power of the MR analysis. Statistical power in MR is directly influenced by the strength and number of genetic instruments, as well as the proportion of variance in the exposure they explain [15]. Although the use of stringent selection criteria ensured strong associations between the selected SNPs and sleep duration—thereby reducing the risk of bias from weak instruments and facilitating the identification of independent variants—low SNP count can still result in limited explanatory variance. This, in turn, may lead to imprecise causal estimates [78]. Such limitations reduce the likelihood of detecting true causal effects and may produce wider confidence intervals and attenuation of effect sizes toward the null. Consequently, the null findings for long sleep duration should be interpreted with caution, as they may reflect limited instrument strength rather than a true absence of causal association. Third, the MR analyses were restricted to individuals of European ancestry, limiting the generalisability of the findings to other populations. This is a notable limitation given the significant differences in sleep patterns, prevalence of sleep disorders, and dementia risk profiles across ethnic groups. For example, research suggests that individuals of African, Asian, and Hispanic ancestry may have different circadian rhythms, sleep duration distributions, and susceptibility to sleep-related health outcomes compared with individuals of European ancestry [79]. Similarly, genetic architecture (including allele frequencies and patterns of linkage disequilibrium) may differ significantly across populations, which may alter the strength or direction of causal relationships [80]. If causal pathways vary by descent, clinical recommendations based on European data alone may not be used in all populations. Future studies should consider validation in cohorts of diverse ancestry. Fourth, horizontal pleiotropy cannot be completely ruled out. In order to increase the robustness of the findings as well as to reduce the potential impact of horizontal pleiotropy, this study supplemented the main IVW analysis with MR-Egger and WME methods. The WME method provides valid causal estimates even when up to 50% of the IVs are invalid. The convergence of the results derived from the three methods enhances confidence in the validity of the findings and reduces the likelihood that the observed associations are attributable to pleiotropic bias. In addition, MR-Egger regression can detect directional pleiotropy through its intercept term, which reflects the average pleiotropic effect across IVs. The results suggest that the MR-Egger intercept is not statistically significant, suggesting limited evidence of horizontal pleiotropy. In addition, the MR-PRESSO detects potential pleiotropy by identifying and correcting for anomalous variation that may overly affect the results. The absence of significant outliers or large distortions in the MR-PRESSO supports the reliability of the estimates. In addition, there may be residual effects and vertical pleiotropy

in this study; in particular, insomnia may overlap phenotypically with self-reported short sleep duration. Finally, the MR study relied on GWAS data, which provide valuable but partial insights into the heritability of complex traits such as sleep duration. The polygenic structure of cognitive outcomes means that even large-scale GWAS can only capture part of the overall phenotypic variation [28,81]. Thus, the genetic tools used in MR studies explain only a limited portion of the complex causal pathways that influence cognition and dementia risk. An individual's genetic risk may only manifest itself in a particular context. For example, individuals with the same risk genes perform differently in cognitive development, in part due to differences in education or nutrition. MR analyses typically assume homogenous genetic effects, which may underestimate the true complexity of gene-environment interactions. Nonetheless, despite these limitations, MR offers a powerful tool for assessing causality in observational research. By leveraging genetic instruments, MR enables causal inference even in the face of incomplete heritability information. Taken together, these findings provide important evidence for the causal role of sleep traits in cognitive decline and dementia, while also highlighting the need for cautious interpretation and further validation in diverse populations and methodological frameworks.

## 5 Conclusions

In this systematic MR study, self-reported short sleep duration was causally associated with cognitive decline and self-reported insomnia was associated with increased risk of AD. However, due to the limited number of SNPs available as instruments for long sleep duration, the evidence for a causal relationship between long sleep duration and cognitive function remains inconclusive and warrants further investigation.

## Supporting information

**S1 Table. Detailed information for cognitive function tests and dementia-related outcomes.** SNPs analysis of short sleep duration. SNPs analysis of insomnia. SNPs analysis of long sleep duration. MR-Egger test for MR analysis of sleep phenotypes and cognitive functions. Cochran's Q statistics test for MR analysis of sleep phenotypes and cognitive functions. The results of MR analyses using the Outlier-corrected method in MR-PRESSO with the results of the original MR analyses. The no. of SNPs determined in this MR study.
(DOCX)

**S1 Fig. Scatterplot of Mendelian randomisation analysis.** Funnel plots for MR analysis of sleep phenotypes and cognitive functions. Leave-one-out analysis for MR analysis of sleep phenotypes and cognitive functions.
(DOCX)

## Acknowledgments

Thanks to the investigators who performed GWAS analyses and summarised the data available in this study.

## Author contributions

**Conceptualization:** Yunyun Guo.

**Methodology:** Yunyun Guo.

**Writing – original draft:** Yunyun Guo.

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
