## [Decision Letter · Decision Letter 0]

31 Mar 2025

Dear Dr. GUO,

Thank you for submitting your manuscript to PLOS ONE. After careful consideration, we feel that it has merit but does not fully meet PLOS ONE’s publication criteria as it currently stands. Therefore, we invite you to submit a revised version of the manuscript that addresses the points raised during the review process.

We look forward to receiving your revised manuscript.

Kind regards,

Zhengrui Li

Academic Editor

PLOS ONE

Journal Requirements:

3. Please ensure that you refer to Figures 3-6 in your text as, if accepted, production will need this reference to link the reader to the figure.

Reviewers' comments:

Reviewer's Responses to Questions

**Comments to the Author**

1. Is the manuscript technically sound, and do the data support the conclusions?

Reviewer #1: Yes

Reviewer #2: Yes

Reviewer #3: Partly

Reviewer #4: Yes

2. Has the statistical analysis been performed appropriately and rigorously?

Reviewer #1: Yes

Reviewer #2: Yes

Reviewer #3: Yes

Reviewer #4: Yes

3. Have the authors made all data underlying the findings in their manuscript fully available?

Reviewer #1: Yes

Reviewer #2: Yes

Reviewer #3: Yes

Reviewer #4: Yes

4. Is the manuscript presented in an intelligible fashion and written in standard English?

Reviewer #1: Yes

Reviewer #2: Yes

Reviewer #3: Yes

Reviewer #4: Yes

Reviewer #1: The manuscript aims to determine whether self‐reported sleep phenotypes—specifically short sleep duration, long sleep duration, and insomnia—are causally linked to cognitive function and dementia risk using a two-sample Mendelian randomization (MR) design. Overall, the study is methodologically sound and addresses an important public health question. The selection of genetic instruments from well-powered GWAS datasets and the use of multiple MR methods are notable strengths.

However, some issues should be addressed before publication:

1. Figures and Results Description:

• The main text would benefit from more detailed commentary on the results shown in Figures 3, 4, 5, and 6. Clearly explain how the associations differ among short sleep duration, long sleep duration, and insomnia across the various cognitive outcomes.

• Please check that all the Figures are referenced in the main text, as this improves readability.

2. Interpretation of Sleep Phenotypes:

• Provide additional insights into why short sleep duration appears to have a stronger association with certain cognitive measures (e.g., fluid intelligence) compared to insomnia, and discuss how these different sleep measures may influence cognitive outcomes via distinct biological pathways.

• Address the non-significant findings for long sleep duration. In particular, comment on how the limited number of SNPs (6–7) used as instruments for long sleep duration might affect statistical power and precision.

3. Literature Context and Novelty Claims:

• The manuscript already provides a clear discussion of what is novel about your study—such as the inclusion of multiple cognitive outcomes and the unique methodological details—when comparing your results with those of Henry et al. (2019). To strengthen the literature context further, I recommend incorporating a discussion of Wang et al. (2023) (J. Evid. Based Med.), which has examined similar associations between sleep traits and cognitive outcomes. Acknowledging Wang et al. (2023) will help situate your findings within the broader MR literature and clarify the added value of your study.

• This revision will ensure that the claim of novelty is appropriately nuanced.

4. Overall Discussion Flow:

• The discussion section appears somewhat repetitive, particularly regarding the reiteration of methodological assumptions and novelty claims. I suggest reorganizing the discussion to improve flow by consolidating these points. A more concise presentation of the core MR assumptions (relevance, independence, and exclusion restriction) and a focused discussion on the study’s limitations (e.g., reliance on self-reported sleep measures and restriction to European ancestry) would enhance readability. If these issues are already covered in other sections, consider removing or rephrasing redundant passages (for example, it is stated twice that "this is the first MR study to investigate causal associations between long sleep duration, short sleep duration, insomnia and multiple cognitive outcomes").

5. Minor Editorial Corrections:

• Page 16. Please revise the incomplete sentence: “Further screening was carried out according to the instrumental variable screening conditions and the final results are presented in Supplement II to IV and .” Clarify what supplement is missing or restructure the sentence for clarity.

• Review the manuscript for additional typographical and grammatical errors to improve clarity and readability. For instance, in the sentence on page 16 discussing the MR-Egger intercept test, consider revising to: "According to the MR-Egger intercept test, there was no evidence of horizontal pleiotropy (p > 0.05), indicating that the data were suitable for MR analysis."

Reviewer #2: This is a well-written manuscript contributing to the understanding of the relation between self-reported sleep duration and cognitive function. The study's design is robust and methodologically sound, and its results are well presented based on the following:

1. The author has clearly stated the aim of this study, which is based on a thorough evaluation of the contradictory findings of previous observational studies of the relation between sleep duration and cognitive decline, using different adjustment models.

2. The study used large-scale GWAS studies with large sample sizes, which enhances its statistical power.

3. Although the author relied on data extracted from multiple GWAS studies, all these studies include individuals of European ancestry, which limits bias due to population differences.

4. The instrumental variables, SNPs, used in the study were thoroughly selected after considering linkage disequilibrium, strand orientation consistency, and checking their association with other genomic traits to exclude confounding factors.

5. Compared to previous literature which relied on the use of Alzheimer Disease as a cognitive outcome, this study is comprehensive. It is based on the evaluation of a broad range of cognitive outcomes, including 4 types of dementia and 7 cognitive tests. Each test comprises distinct subtests, scoring mechanisms, and participant demographics.

6. The study uses several MR methods; IVW supplemented with MR-Egger regression and the Weighted Median approach, followed by multiple sensitivity analyses. The use of these methods is well presented in the Methodology, however, the explanation of how they support the robustness of the findings and address potential pleiotropy could be elaborated in the discussion.

7. The study relies on Mendelian Randomization analysis to find causal relationships, since it alleviates bias from confounding and reverse causation. In line of this argument, the recent study on the Relationship of Sleep Duration on Cognitive Function Among the Elderly (Qiu et al., 2024) is worth mentioning. It combines data from the National Health and Nutrition Examination Survey (NHANES) and Mendelian randomization methods. The findings of this article could be also discussed in the discussion part.

8. The fact that genetic predisposition may only explain part of the variance related to cognitive outcomes is acknowledged in the discussion but could be further elaborated.

9. While the author discusses the effect of sleep duration on Aβ deposition and clearance, the discussion could acknowledge potential biological mechanisms that explain the causal relationships, such as the relation between sleep quality, neuroinflammation and cognitive health.

Reviewer #3: This study may exhibit several limitations:

1- The genetic variations in the SNPs determined by this study were already related to loss mental abilities by dementia or Alzheimer, therefor, the reliance to self-reporting duration to measure the cognitive may be not accurate and lead to biased results.

2- The study restricted the randomized analysis on the European ancestry population. So, the results may not match with variation in another ancestry populations.

3- The genetic variations influence sleep duration is unclear and affected by several environmental factors, sex, age …etc. therefor it’s not enough to be decisive results to link the cognitive with sleep duration.

Reviewer #4: Similar studies were published earlier concluding the same outcomes that prescribes a normal sleep duration of 7-8 hours in a day results in healthy cognitive function. However, this study has the uniqueness of implementing new MR strategy with a robust workflow that rules out some bias present in the previous studies especially with respect to the long sleep duration. Limitation of the study were discussed as well.

**Do you want your identity to be public for this peer review?** For information about this choice, including consent withdrawal, please see our Privacy Policy

Reviewer #1: No

Reviewer #2: No

Reviewer #3: No

Reviewer #4: No

---

## [Author Response · Author response to Decision Letter 1]

14 May 2025

Mendelian Randomization Study of Self-reported Long sleep duration, Short sleep duration, and Insomnia and Cognitive Function

Thank you for the opportunity to revise and resubmit this manuscript. The comments and suggestions provided by reviewers have been very helpful in guiding the revision. The manuscript have been revised thoroughly in accordance with recommendations of the reviewers. A detailed summary of the revisions was provided in the accompanying memo. I believe that the manuscript has improved significantly as a result of the review process, and hope that the revised study is suitable for a potential publication in PLOS One.

ACADEMIC EDITOR

1.Please ensure that your manuscript meets PLOS ONE's style requirements, including those for file naming.

I have revised the manuscript to conform to PLOS ONE’s style requirements, including formatting and file naming, following the templates provided.

2.PLOS requires an ORCID iD for the corresponding author in Editorial Manager on papers submitted after December 6th, 2016. Please ensure that you have an ORCID iD and that it is validated in Editorial Manager. To do this, go to ‘Update my Information’ (in the upper left-hand corner of the main menu), and click on the Fetch/Validate link next to the ORCID field. This will take you to the ORCID site and allow you to create a new iD or authenticate a pre-existing iD in Editorial Manager.

No problem! Will do.

3. Please ensure that you refer to Figures 3-6 in your text as, if accepted, production will need this reference to link the reader to the figure.

I have ensured that all figures, including Figures 3–6, are now properly referenced in the main text.

Captions for all Supporting Information files have been added to S1 Fig and S1 Table, and in-text citations have been updated accordingly.

5.Please review your reference list to ensure that it is complete and correct. If you have cited papers that have been retracted, please include the rationale for doing so in the manuscript text, or remove these references and replace them with relevant current references. Any changes to the reference list should be mentioned in the rebuttal letter that accompanies your revised manuscript. If you need to cite a retracted article, indicate the article’s retracted status in the References list and also include a citation and full reference for the retraction notice.

I have updated the reference list to ensure its accuracy and completeness. No retracted articles are cited. Any references that were added or modified are described below.

REVIEWER 1

The manuscript aims to determine whether self‐reported sleep phenotypes—specifically short sleep duration, long sleep duration, and insomnia—are causally linked to cognitive function and dementia risk using a two-sample Mendelian randomization (MR) design. Overall, the study is methodologically sound and addresses an important public health question. The selection of genetic instruments from well-powered GWAS datasets and the use of multiple MR methods are notable strengths. However, some issues should be addressed before publication:

Thank you for your thorough review of our manuscript and positive and encouraging feedback. I appreciate the time that you have put in to it. I also appreciate the concerns you have raised. To that end, I have addressed each of your concerns, and I believe that the revised manuscript is stronger as a result. Below are responses to each of your comments. I also added a list of references at the end of your comments. Revisions within the manuscript are highlighted in blue.

1. Figures and Results Description:

• The main text would benefit from more detailed commentary on the results shown in Figures 3, 4, 5, and 6. Clearly explain how the associations differ among short sleep duration, long sleep duration, and insomnia across the various cognitive outcomes.

Thank you for this suggestion. I have added more detailed commentary on the results presented in Figures 3, 4, 5, and 6. This includes highlighting patterns of directionality, magnitude of effect sizes, and statistical significance. The details are as follows

“The findings suggest that short sleep duration may adversely affect multiple domains of cognitive function. MR analysis using an IVW method showed that short sleep duration showed a harmful impact on cognitive performance (beta = –0.15, 95% CI: –0.27 to –0.02, P = 0.02), which serve as a potential test for identifying people in the community with undiagnosed dementia, and their decline reflects a significant reduction in overall cognitive function. Similarly, there was a strong negative correlation between short sleep duration and FIS (beta = –0.38; 95% CI: –0.65 to –0.11; P = 0.006, IVW). The FIS is a physiologically based measure of cognitive ability related to arithmetic speed and reasoning. In terms of memory function, short sleep duration was linked to poorer memory performance (beta = –0.10, 95% CI: –0.20 to –0.0005; P = 0.04, IVW), further underscoring the broad cognitive impact of inadequate sleep (Figure 3).

As shown in Figure 3, performance on the TM test, which assesses executive function and the ability to switch attention, was also negatively affected. Specifically, short sleep duration was associated with a longer interval between steps in the Trail 2 path (beta = 0.11, 95% CI: 0.01 to 0.21, P = 0.03, IVW) and a longer total duration to complete the Trail 2 path (beta = 0.11, 95% CI: 0.002 to 0.22, P = 0.04, IVW), suggesting impairments in attention and task-switching capabilities.

There was insufficient evidence to support a causal relationship between insomnia and performance on cognitive tests, including FIS (Figure 4). In addition, insomnia was causally associated with AD, with a significant odds ratio of 1.13 (95% CI: 1.02 to 1.24, P = 0.02, IVW) (Figure 6). However, there was insufficient evidence to support a causal relationship between long sleep duration and every cognitive-related outcome (P > 0.05 for all) (Figure 5 and Figure 6). ”

Please refer to 3.2 Primary analysis.

• Please check that all the Figures are referenced in the main text, as this improves readability.

I have also carefully checked the manuscript to ensure that all figures are explicitly referenced in the main text at appropriate points, in order to improve readability and coherence.

Kindly refer to 3.2 Primary analysis.

2. Interpretation of Sleep Phenotypes:

• Provide additional insights into why short sleep duration appears to have a stronger association with certain cognitive measures (e.g., fluid intelligence) compared to insomnia, and discuss how these different sleep measures may influence cognitive outcomes via distinct biological pathways.

I appreciate this insightful suggestion regarding the interpretation of the associations. Indeed, short sleep duration seems to be more strongly associated with certain cognitive indicators (e.g., fluid intelligence) than insomnia. This may be due to differences in the definitions of short sleep duration and insomnia. In this study, short sleep duration was defined as 6 hours or less of sleep per 24 hours. Shorter sleep duration implies a reduction in slow wave sleep (SWS) and rapid eye movement (REM) sleep.The emergence and integrity of SWS are strongly associated with fact-dependent situational memory capacity (1). After controlling for the effects of demographic factors, studies have found an association between SWS sleep and information processing speed; meanwhile, REM sleep significantly affects an individual's executive function performance (2). Insomnia is defined as difficulty falling asleep at night or waking up in the middle of the night. Thus, insomniacs may still obtain a normal total sleep duration, albeit with poorer sleep quality. Alternatively, insomniacs may be more conscious of this compensation through planned daytime naps, but chronically sleep-deprived individuals tend to underestimate their cognitive deficits. During planned daytime naps, individuals may not only compensate for REM sleep, but also compensate for reduced SWS by prolonging REM sleep, thus demonstrating resilience to SWS deficits. From this perspective, some of the functions of SWS sleep may be redistributed during REM sleep stages or compensated to some extent by REM sleep-related processes (2). Second, sleep duration in this study was supported by accelerometer estimates. Insomnia relies on subjective self-reports, and subjective symptoms are influenced by mood, anxiety, and personal distress, which complicates its connection to objective cognition. In contrast to disease diagnosis, fluid intelligence emphasises speed of information processing, ability to reason abstractly and executive function. Thus, fluid intelligence scores better capture the changes in cognitive functioning that occur as a result of reduced SWS and REM due to short sleep duration. The above has been added to the seventh paragraph of the Discussion.

In addition to the fact that sleep duration affects cognitive function by influencing SWS and REM emergence and integrity, I have added the biological mechanisms by which sleep disruption or fragmentation due to insomnia affects cognitive health by influencing neuroinflammation and cerebral lymphatic system clearance. It has been added as follows:

“Beyond sleep duration, growing evidence suggests that sleep quality plays a critical role in brain health through its effects on neuroinflammation and glymphatic clearance. Disrupted or fragmented sleep has been shown to increase systemic and central nervous system levels of pro-inflammatory cytokines, such as interleukin-6 (IL-6) and tumor necrosis factor-alpha (TNF-α), which can exacerbate neuroinflammatory processes and contribute to neurodegeneration (48). Inadequate or poor-quality sleep may also impair the function of the glymphatic system—a waste clearance pathway active during sleep—thereby reducing the removal efficiency of amyloid-β and other neurotoxic proteins from the brain (47). Chronic neuroinflammation has been implicated in synaptic dysfunction, tau pathology, and accelerated cognitive decline, suggesting that sleep quality is a key modifiable factor in the pathway linking sleep disturbance to AD (49). These biological mechanisms highlight the importance of considering both the duration and quality of sleep when evaluating its causal role in cognitive aging and neurodegenerative disease.”

Please refer to the third paragraph of the Discussion.

• Address the non-significant findings for long sleep duration. In particular, comment on how the limited number of SNPs (6–7) used as instruments for long sleep duration might affect statistical power and precision.

Thanks for pointing this out. This study found no evidence of statistically significant support for a causal association between long sleep duration and cognitive function. The limited number of SNPs used as instrumental variables for long sleep duration likely constrained the statistical power of our MR analysis. Statistical power in MR is directly influenced by the strength and number of genetic instruments, as well as the proportion of variance in the exposure they explain (3). The rigorous selection criteria chosen for this study ensured strong associations between the selected SNPs and sleep duration, thus avoiding bias from weak instrumental variables and helping to select independent genetic variants. However, when only a small number of SNPs are available, the explained variance tends to be low, which can lead to imprecise causal estimates (4). This not only reduces the likelihood of detecting true causal effects but may also result in wider confidence intervals and attenuation of effect sizes toward the null. Consequently, the null findings for long sleep duration should be interpreted with caution, as they may reflect limited instrument strength rather than a true absence of causal association.

The above has been incorporated into the Discussion section; please refer to the second second limitation point within the Discussion.

3. Literature Context and Novelty Claims:

• The manuscript already provides a clear discussion of what is novel about your study—such as the inclusion of multiple cognitive outcomes and the unique methodological details—when comparing your results with those of Henry et al. (2019). To strengthen the literature context further, I recommend incorporating a discussion of Wang et al. (2023) (J. Evid. Based Med.), which has examined similar associations between sleep traits and cognitive outcomes. Acknowledging Wang et al. (2023) will help situate your findings within the broader MR literature and clarify the added value of your study.

• This revision will ensure that the claim of novelty is appropriately nuanced.

Sure thing! Done! A study by Wang et al. (2023) (J. Evid. Based Med.) has been added and is discussed further along with other relevant MR analyses. The details are as follows

“The MR study by Wang et al. explored the potential causal relationship between sleep characteristics including sleep duration, insomnia and cognitive impairment (58). This study reported a negative correlation between self-reported sleep duration and reaction time, suggesting an adverse effect of short sleep duration on cognitive performance. This study did not find sufficient evidence to support a causal relationship between insomnia and cognitive function (58).”

“Five previous MR studies have brought enlightening evidence (57–61). Based on previous MR analyses, the present study further explored the causal relationship between sleep duration and cognitive functioning by further categorising sleep duration: long sleep duration and short sleep duration. In addition, the present study used insomnia GWAS data based on a larger sample size for exploration. A review of previous MR analyses on insomnia and cognitive functioning found that the study by Wang et al. used insomnia GWAS data from 386,533 individuals of European ancestry recruited from 22 assessment centres in the UK (58); the study by Huang et al. included insomnia GWAS data from 237,627 individuals of European ancestry in the UK (61). The present study used combined data collected by UK Biobank (n = 386,533 individuals) and 23andMe (n = 944,477 individuals) for a final sample size of up to 1,331,010 individuals of European ancestry. Insomnia studies with different sample sizes and study sites may yield different conclusions. On the other hand, Qiu et al.'s MR analyses considered only one test of cognitive function (59). Henry et al. judged cognitive function only by visual memory tests, reaction time and dementia or not (57). However, the manifestations of cognitive decline are complex. It is difficult to fully reflect changes in cognitive functioning and the presence of dementia using only one or two tests. Compared with the study of Henry et al, Wang et al included more cognitive tests such as cognitive performance, fluid intelligence (58). Based on previous studies, the present study added more cognitive function tests (seven cognitive tests including memory test, reaction time cognitive performance and fluid intelligence). At the same time, previous MR studies did not provide more evidence about sleep characteristics and dementia subtypes. Qiu et al.'s MR analysis did not consider any dementia and its subtypes (59); Henry et al., Yuan et al. and Huang et al. all focused only on AD, the most common form of dementia (57, 60, 61); and Wang et al.'s focus fell on AD-related progression scores and changes in brain structure (58). The present study extends beyond AD to more common subtypes of dementia, namely vascular dementia, Lewy body dementia, and frontotemporal dementia.”

Kindly refer to the fifth and sixth paragraphs of the Discussion.

4. Overall Discussion Flow:

• The discussion section appears somewhat repetitive, particularly regarding the reiteration of me

---

## [Decision Letter · Decision Letter 1]

2 Jun 2025

Dear Dr. GUO,

Thank you for submitting your manuscript to PLOS ONE. After careful consideration, we feel that it has merit but does not fully meet PLOS ONE’s publication criteria as it currently stands. Therefore, we invite you to submit a revised version of the manuscript that addresses the points raised during the review process.

We look forward to receiving your revised manuscript.

Kind regards,

Zhengrui Li

Academic Editor

PLOS ONE

**Journal Requirements:**

Reviewers' comments:

Reviewer's Responses to Questions

**Comments to the Author**

Reviewer #1: All comments have been addressed

Reviewer #5: All comments have been addressed

Reviewer #6: (No Response)

2. Is the manuscript technically sound, and do the data support the conclusions?

Reviewer #1: Yes

Reviewer #5: Yes

Reviewer #6: Partly

3. Has the statistical analysis been performed appropriately and rigorously?

Reviewer #1: Yes

Reviewer #5: Yes

Reviewer #6: Yes

4. Have the authors made all data underlying the findings in their manuscript fully available?

Reviewer #1: Yes

Reviewer #5: Yes

Reviewer #6: Yes

5. Is the manuscript presented in an intelligible fashion and written in standard English?

Reviewer #1: Yes

Reviewer #5: Yes

Reviewer #6: Yes

**Reviewer #1:**  Thank you for your thorough and thoughtful responses to my initial review. I am pleased to confirm that all of my concerns have been fully addressed in the revised manuscript.

The additions to the Results and Discussion sections offer a much clearer interpretation of the data, particularly regarding the distinct associations of short sleep duration, insomnia, and long sleep duration with cognitive outcomes. The clarification of potential biological mechanisms, especially in terms of sleep architecture and inflammatory pathways, adds valuable context to the findings. The inclusion of relevant literature (e.g., Wang et al., 2023) and the improved discussion flow also strengthen the overall narrative and support the manuscript’s contribution to the Mendelian randomization literature.

Figures have been referenced appropriately, editorial issues have been resolved, and the limitations of the MR design—including instrument strength and sleep phenotype definitions—are now well-articulated.

I have no further suggestions for revision. The manuscript is methodologically sound, clearly written, and meets the standards for publication. I recommend acceptance in its current form.

**Reviewer #5: ** This is a well-written manuscript exploring the causal associations between sleep hours and cognitive functions using Mendelian randomization. The paper has clearly stated how the statistical analyses were conducted and summarized the strength and limitations of the methods.

**Reviewer #6:**  This study investigates whether sleep patterns, specifically short or long sleep duration and insomnia, causally influence cognitive performance and dementia risk. The researchers employed Mendelian Randomization to leverage genetic variants as instrumental variables for testing causal relationships. They found that genetically predicted short sleep duration impairs cognitive performance across multiple domains including memory and attention. Insomnia showed a causal association with Alzheimer's disease risk but did not affect general cognitive test performance. Long sleep duration effects remain inconclusive due to insufficient genetic instruments.

Major Comments

1. Insufficient Statistical Power for Long Sleep Duration Analysis: Genetic instrument for long sleep duration contains only 7 SNPs, severely limiting statistical power. The abstract and discussion imply absence of causal effects rather than acknowledging inadequate power to detect them. Revise language throughout to clarify that findings reflect limited instrumental strength rather than definitive evidence against causal effects.

2. Disconnect Between Insomnia Effects on AD vs. Cognitive Performance: Your finding that insomnia causally increases Alzheimer's risk while showing no effect on cognitive test scores requires deeper exploration. The brief discussion doesn't adequately address this apparent contradiction. Expand discussion of potential mechanisms underlying this dissociation, considering whether UK Biobank cognitive assessments may lack sensitivity to detect subtle neurodegenerative changes that manifest clinically as AD.

3. Self-Report Bias in Sleep Phenotypes: Reliance on self-reported sleep data introduces potential misclassification bias that could attenuate true causal estimates. Subjective sleep perception varies considerably between individuals, particularly for insomnia symptoms. Discuss how measurement error in self-reported exposures might bias results toward the null and emphasize the need for future studies incorporating objective sleep measurements.

4. Mechanistic Discussion Lacks Domain Specificity: Your biological discussion mentions relevant pathways but fails to connect these mechanisms to your domain-specific cognitive findings. Link specific sleep-related mechanisms to affected cognitive domains, discuss how REM sleep disruption impacts attention and executive function while slow-wave sleep disruption affects memory consolidation.

5. Overstated Novelty Claims: The introduction presents this as the first MR study of sleep-cognition relationships, but you later cite several relevant prior MR studies. This inconsistent framing undermines credibility. Refine novelty claims to highlight genuine contributions: comprehensive cognitive phenotyping, inclusion of multiple dementia subtypes, and largest sample size for insomnia analysis.

6. Limited Generalizability Beyond European Ancestry: Restricting analyses to European ancestry populations limits generalizability, particularly given known ancestry differences in sleep patterns and dementia risk profiles. Explicitly discuss this limitation's public health implications and emphasize the need for replication in diverse populations before clinical translation.

Minor Comments:

1. Sensitivity analyses are comprehensive and well-executed, strengthening confidence in the primary findings.

2. Effect sizes for some cognitive outcomes, while statistically significant, appear small. Providing clinical context or benchmarks would help readers assess practical significance.

3. Consider adding brief justification for your sleep duration thresholds (<6h, >9h) in the Methods section.

4. Discussion flow has improved but could benefit from eliminating remaining redundancy between cognitive and dementia sections.

**Do you want your identity to be public for this peer review?** For information about this choice, including consent withdrawal, please see our Privacy Policy

Reviewer #1: No

Reviewer #5: No

Reviewer #6: No

---

## [Author Response · Author response to Decision Letter 2]

9 Jul 2025

PONE-D-24-46332R2

Mendelian Randomization Study of Self-reported Long sleep duration, Short sleep duration, and Insomnia and Cognitive Function

Thank you for the opportunity to revise and resubmit the manuscript. I appreciate the reviewers’ constructive feedback, which has helped us improve the clarity and quality of the work. I have addressed all comments carefully and outlined the responses in the accompanying revision memo. I hope the revised manuscript meets the standards for publication in PLOS ONE.

Editor

Thank you for your response. We still require accession numbers so that interested researchers may search up your exact datasets that have been used. Currently, the links do not lead directly to the datasets and instead lead to the homepages of the websites.

I have removed the general website homepage links from the manuscript—the sentence "Genetic data pertaining to cognitive function were sourced from the UK Biobank (https://www.ukbiobank.ac.uk) and the FinnGen consortium (https://www.finngen.fi/fi)" has been deleted. In addition, I have added direct links to the specific datasets used in S1 Table, so that interested researchers may easily access the exact data.

Reviewer #1: Thank you for your thorough and thoughtful responses to my initial review. I am pleased to confirm that all of my concerns have been fully addressed in the revised manuscript.

The additions to the Results and Discussion sections offer a much clearer interpretation of the data, particularly regarding the distinct associations of short sleep duration, insomnia, and long sleep duration with cognitive outcomes. The clarification of potential biological mechanisms, especially in terms of sleep architecture and inflammatory pathways, adds valuable context to the findings. The inclusion of relevant literature (e.g., Wang et al., 2023) and the improved discussion flow also strengthen the overall narrative and support the manuscript’s contribution to the Mendelian randomization literature.

Figures have been referenced appropriately, editorial issues have been resolved, and the limitations of the MR design—including instrument strength and sleep phenotype definitions—are now well-articulated.

I have no further suggestions for revision. The manuscript is methodologically sound, clearly written, and meets the standards for publication. I recommend acceptance in its current form.

Thank you for the positive and encouraging feedback. It is reassuring to know that the revisions effectively addressed the previous concerns.

Thank you again for the thoughtful review and recommendation for acceptance.

Reviewer #5: This is a well-written manuscript exploring the causal associations between sleep hours and cognitive functions using Mendelian randomization. The paper has clearly stated how the statistical analyses were conducted and summarized the strength and limitations of the methods.

Thank you for the positive feedback

Reviewer #6: This study investigates whether sleep patterns, specifically short or long sleep duration and insomnia, causally influence cognitive performance and dementia risk. The researchers employed Mendelian Randomization to leverage genetic variants as instrumental variables for testing causal relationships. They found that genetically predicted short sleep duration impairs cognitive performance across multiple domains including memory and attention. Insomnia showed a causal association with Alzheimer's disease risk but did not affect general cognitive test performance. Long sleep duration effects remain inconclusive due to insufficient genetic instruments.

Thank you for your thorough and professional review of the manuscript. As you are concerned, there are several problems that need to be addressed. I have addressed and explained each of your concerns, and I believe that the revised manuscript is stronger as a result. Below are responses to each of your comments. Revisions within the manuscript are highlighted in blue.

Major Comments

1. Insufficient Statistical Power for Long Sleep Duration Analysis: Genetic instrument for long sleep duration contains only 7 SNPs, severely limiting statistical power. The abstract and discussion imply absence of causal effects rather than acknowledging inadequate power to detect them. Revise language throughout to clarify that findings reflect limited instrumental strength rather than definitive evidence against causal effects.

Thank you for highlighting the important issue of limited statistical power in the analysis of long sleep duration. The concern regarding the small number of SNPs (n=7) used as the genetic instrument is fully acknowledged. To address this, the language in the Abstract, Results, and Discussion sections has been revised to clarify that the null findings for long sleep duration should be interpreted with caution, reflecting the limited strength of the instrument rather than definitive evidence against a causal relationship.

In the Abstract, the corresponding section is written as:

“However, due to the limited number of SNPs (n = 7) available as instruments for long sleep duration, there was no strong evidence to support a causal effect of long sleep duration on cognitive outcomes.”

“The evidence of causality between long sleep duration and cognitive function requires further investigation.”

In the Results, the corresponding section is written as:

“Considering the limited number of SNPs, the causal relationship between long sleep duration and cognitive function still needs further investigation.”

In the Discussion, the corresponding section is written as:

“The evidence of causality between long sleep duration and cognitive function remains unclear and requires further investigation.”

“Considering the limited number of SNPs (n = 7) available as instruments for long sleep duration, there was no strong evidence to support a causal effect of long sleep duration on cognitive outcomes.”

In the Conclusions, the corresponding section is written as:

“However, due to the limited number of SNPs available as instruments for long sleep duration, the evidence for a causal relationship between long sleep duration and cognitive function remains inconclusive and warrants further investigation.”

Please refer to the last sentence of the results and the second sentence of the discussion in the Abstract, the last sentence of 3.2 Primary analysis in the Results, the last sentence of the first paragraph and the first sentence of the fourth paragraph in the Discussion and the last sentence in the Conclusions.

2. Disconnect Between Insomnia Effects on AD vs. Cognitive Performance: Your finding that insomnia causally increases Alzheimer's risk while showing no effect on cognitive test scores requires deeper exploration. The brief discussion doesn't adequately address this apparent contradiction. Expand discussion of potential mechanisms underlying this dissociation, considering whether UK Biobank cognitive assessments may lack sensitivity to detect subtle neurodegenerative changes that manifest clinically as AD.

Thank you for highlighting the important point. The Discussion section has been expanded to explore potential mechanisms underlying this discrepancy. Specifically, I wrote:

“The present study suggests a close causal link between insomnia and AD, rather than cognitive test scores. There are several plausible explanations for this dissociation. Firstly, cognitive assessment methods provided by the UK Biobank may not readily detect cognitive decline in the preclinical stages of AD. These assessment methods, although suitable for large-scale population studies, may lack the sensitivity to capture subtle or domain-specific cognitive deficits, such as language deficits, disorientation disorders, which are often among the areas susceptible to AD (24, 40). Thus, individuals with early neurodegenerative conditions may still perform normally on these relatively crude tests. Secondly, insomnia may increase long-term risk for AD through pathways that do not lead to immediately measurable cognitive deficits. For example, insomnia is associated with impaired clearance of amyloid β-protein and tau protein by the lymphatic system, systemic inflammation, and neuroendocrine dysregulation, all of which are implicated in the pathogenesis of AD but may precede observable changes in cognitive performance by years or even decades (44, 46, 49). Thirdly, individuals with chronic insomnia may also develop compensatory mechanisms or cognitive reserves that temporarily buffer measurable performance deficits, thereby delaying the emergence of impairments on cognitive tests (64). Finally, in the GWAS included in this study, the diagnosis of AD captures a clinical endpoint, whereas the cognitive performance indicators in the UK Biobank reflect subclinical characteristics. The lack of association between insomnia and cognitive test scores does not necessarily imply a lack of cognitive impact, but should reflect the complexity and time lag between sleep disorders and cognitive decline.”

Kindly refer to the ninth paragraph of the Discussion section.

3. Self-Report Bias in Sleep Phenotypes: Reliance on self-reported sleep data introduces potential misclassification bias that could attenuate true causal estimates. Subjective sleep perception varies considerably between individuals, particularly for insomnia symptoms. Discuss how measurement error in self-reported exposures might bias results toward the null and emphasize the need for future studies incorporating objective sleep measurements.

Thanks for pointing this out. To address this concern, I have revised the manuscript to discuss the implications of using self-reported exposures and emphasized the need for future studies using objective sleep measures. The details are as follows:

“First, self-reported sleep duration or insomnia symptoms may introduce a potential misclassification bias due to memory and reporting errors, which can lead to biased MR analysis estimates towards zero values and underestimation of true causal effects. This bias mainly occurs in cases where there is no difference in measurement error. In this case, the noise introduced by self-reporting weakens the observed association between the genetic IV and the exposure, thereby reducing the strength of the IV. This dilution effect reduces the proportion of exposure variance explained by the IVs, leading to regression dilution bias. As a result, the estimated causal effect of sleep on cognitive or neurological outcomes is attenuated, making it appear weaker or even zero, despite the fact that there may be a genuine underlying association. This issue is particularly important for subjective sleep phenotypes such as insomnia, as individual perceptions can vary widely depending on factors such as mood, stress, and attentional biases. In addition, an increase in exposure misclassification reduces statistical efficacy and widens standard errors, thereby increasing the likelihood of Type II errors. The absence of a flawless method for measuring sleep duration without disrupting daily routines underscores the challenge in this domain (56). Despite the often robust correlation between actigraphy and polysomnography findings (72), actigraphy tends to overestimate sleep duration, particularly beyond the confines of sleep laboratory environments (72–76). Notably, genetic correlations with sleep duration appear to be consistent across self-reported and accelerometer-based measures (16), which provides some robustness to the use of self-reported data. In addition, international sleep guidelines continue to rely on self-reported sleep duration to formulate public health recommendations (2, 77), which supports their relevance in both clinical and epidemiological contexts. Nonetheless, future research should prioritise the integration of objective sleep measures (e.g., somatokinetic recorders and polysomnography) into large-scale cohort studies to improve measurement accuracy and reduce potential bias in causal inference. ”

Please kindly refer to the eleventh paragraph of the Discussion section, the first point under the limitations.

4.Mechanistic Discussion Lacks Domain Specificity: Your biological discussion mentions relevant pathways but fails to connect these mechanisms to your domain-specific cognitive findings. Link specific sleep-related mechanisms to affected cognitive domains, discuss how REM sleep disruption impacts attention and executive function while slow-wave sleep disruption affects memory consolidation.

Thank you for this insightful comment. I have revised the Discussion section to include domain-specific links between sleep physiology and cognitive functions. I wrote:

“In the current study, the genetic association analyses yielded some interesting results. Short sleep duration seems to be more strongly associated with certain cognitive indicators, such as fluid intelligence, than insomnia. This may be due to differences in the definitions of short sleep duration and insomnia. In this study, short sleep duration was defined as 6 hours or less of sleep per 24 hours. Shorter sleep duration implies a reduction in slow wave sleep (SWS) and rapid eye movement (REM) sleep. These two sleep stages play different roles in cognitive functioning. SWS, characterised by high-amplitude, low-frequency brain waves, is essential for the hippocampal-neocortical dialogue during memory consolidation (62). Diminished SWS impairs the reactivation of newly acquired information from the hippocampus and its transfer to long-term storage in the cortex (62). Thus, SWS is critically associated with declarative memory consolidation and fact-based situational memory (62). It has been shown that SWS disruption is associated with impaired situational memory and slower information processing, especially after accounting for demographic and lifestyle factors (62). In contrast, REM sleep is known to promote synaptic plasticity and activate neural circuits involving the prefrontal cortex and striatum to support emotion regulation, attention, and executive function. REM sleep disruption has been shown to impair sustained attention, working memory, and cognitive flexibility, functions that are critically dependent on the integrity of frontal-striatal circuits (63). The UK Biobank assessment reflects fluid intelligence - which encompasses the ability to process and apply new information, reason abstractly, and perform goal-directed behaviours effectively. Therefore, the observed association between short sleep duration and reduced fluid intelligence scores may reflect cumulative deficits caused by reduced SWS, which affects memory encoding and consolidation, and reduced REM sleep, which affects attention and executive functioning. ”

“Insomnia is defined as difficulty falling asleep at night or waking up in the middle of the night. Thus, insomniacs may still obtain a normal total sleep duration, albeit with poorer sleep quality. Alternatively, insomniacs may be more conscious of this compensation through planned daytime naps, but chronically sleep-deprived individuals tend to underestimate their cognitive deficits. During planned daytime naps, individuals may not only compensate for REM sleep, but also compensate for reduced SWS by prolonging REM sleep, thus demonstrating resilience to SWS deficits. From this perspective, some of the functions of SWS sleep may be redistributed during REM sleep stages or compensated to some extent by REM sleep-related processes (63). Second, sleep duration in this study was supported by accelerometer estimates. Insomnia relies on subjective self-reports, and subjective symptoms are influenced by mood, anxiety, and personal distress, which complicates its connection to objective cognition. In contrast to disease diagnosis, fluid intelligence emphasises speed of information processing, ability to reason abstractly and executive function. Thus, fluid intelligence scores better capture the changes in cognitive functioning that occur as a result of reduced SWS and REM due to short sleep duration.”

Kindly refer to the seventh and eighth paragraphs of the Discussion section.

5. Overstated Novelty Claims: The introduction presents this as the first MR study of sleep-cognition relationships, but you later cite se

---

## [Decision Letter · Decision Letter 2]

6 Aug 2025

Mendelian Randomization Study of Self-reported Long sleep duration, Short sleep duration, and Insomnia and Cognitive Function

PONE-D-24-46332R2

Dear Dr. GUO,

We’re pleased to inform you that your manuscript has been judged scientifically suitable for publication and will be formally accepted for publication once it meets all outstanding technical requirements.

Kind regards,

Zhengrui Li

Academic Editor

PLOS ONE

Additional Editor Comments (optional):

Reviewers' comments:

Reviewer's Responses to Questions

**Comments to the Author**

Reviewer #1: All comments have been addressed

Reviewer #5: All comments have been addressed

Reviewer #6: All comments have been addressed

2. Is the manuscript technically sound, and do the data support the conclusions?

Reviewer #1: Yes

Reviewer #5: Yes

Reviewer #6: Yes

3. Has the statistical analysis been performed appropriately and rigorously?

Reviewer #1: Yes

Reviewer #5: Yes

Reviewer #6: Yes

4. Have the authors made all data underlying the findings in their manuscript fully available?

Reviewer #1: Yes

Reviewer #5: Yes

Reviewer #6: Yes

5. Is the manuscript presented in an intelligible fashion and written in standard English?

Reviewer #1: Yes

Reviewer #5: Yes

Reviewer #6: (No Response)

Reviewer #1: I have reviewed the manuscript again, and I confirm that the suggested changes were thoughtfully implemented. I do not have any additional concerns beyond those previously addressed.

Reviewer #5: This is a well-written paper investigating the causal associations between sleep hours and cognitive functions using Mendelian randomization and I don't have additional comments.

Reviewer #6: Good work. Appreciate the detailed response to each comment. The manuscript has been significantly improved.

**Do you want your identity to be public for this peer review?** For information about this choice, including consent withdrawal, please see our Privacy Policy

Reviewer #1: No

Reviewer #5: No

Reviewer #6: No

---

## [Editor Report · Acceptance letter]

PONE-D-24-46332R2

PLOS ONE

Dear Dr. GUO,

I'm pleased to inform you that your manuscript has been deemed suitable for publication in PLOS ONE. Congratulations! Your manuscript is now being handed over to our production team.

Kind regards,

on behalf of

Dr. Zhengrui Li

Academic Editor

PLOS ONE